https://doi.org/10.1038/s41467-021-26328-2　　**OPEN**

# Bayesian log-normal deconvolution for enhanced in silico microdissection of bulk gene expression data

Bárbara Andrade Barbosa[1], Saskia D. van Asten [1,2], Ji Won Oh [3,4], Arantza Farina-Sarasqueta[5], Joanne Verheij[5], Frederike Dijk[5], Hanneke W. M. van Laarhoven[6], Bauke Ylstra [1], Juan J. Garcia Vallejo[2], Mark A. van de Wiel [7 ✉] & Yongsoo Kim [1 ✉]

Deconvolution of bulk gene expression profiles into the cellular components is pivotal to portraying tissue's complex cellular make-up, such as the tumor microenvironment. However, the inherently variable nature of gene expression requires a comprehensive statistical model and reliable prior knowledge of individual cell types that can be obtained from single-cell RNA sequencing. We introduce BLADE (Bayesian Log-normAl Deconvolution), a unified Bayesian framework to estimate both cellular composition and gene expression profiles for each cell type. Unlike previous comprehensive statistical approaches, BLADE can handle > 20 types of cells due to the efficient variational inference. Throughout an intensive evaluation with > 700 simulated and real datasets, BLADE demonstrated enhanced robustness against gene expression variability and better completeness than conventional methods, in particular, to reconstruct gene expression profiles of each cell type. In summary, BLADE is a powerful tool to unravel heterogeneous cellular activity in complex biological systems from standard bulk gene expression data.

[1] Department of Pathology, Cancer Center Amsterdam, Amsterdam UMC, Vrije Universiteit Amsterdam, Amsterdam, the Netherlands. [2] Department of Molecular Cell Biology & Immunology, Amsterdam UMC, Vrije Universiteit Amsterdam, Amsterdam Infection and Immunity Institute, Amsterdam, the Netherlands. [3] Department of Anatomy, School of Medicine, Kyungpook National University, Daegu, South Korea. [4] Bio-Medical Research Institute, Kyungpook National University Hospital, Daegu, South Korea. [5] Department of Pathology, Amsterdam UMC, University of Amsterdam, Amsterdam, the Netherlands. [6] Department of Medical Oncology, Cancer Center Amsterdam, Amsterdam UMC, University of Amsterdam, Amsterdam, the Netherlands. [7] Department of Epidemiology and Data Science, Amsterdam Public Health research institute, Amsterdam UMC, Vrije Universiteit Amsterdam, Amsterdam, the Netherlands. ✉email: mark.vdwiel@amsterdamumc.nl; yo.kim@amsterdamumc.nl

Over the past decade, gene expression profiling has been applied to elucidate the complexity of transcriptional regulation in diverse biological contexts, such as cancer[1,2]. Conventional gene expression profiling, either by RNA sequencing (RNA-seq) or microarrays, captures cumulative gene expression levels of many cells combined. Therefore, it is often referred to as bulk gene expression profiling to distinguish it from the recent single-cell gene expression profiling technologies[3]. In oncology, single-cell RNA sequencing (scRNA-seq) is employed to study cellular heterogeneity within a tumor, composed of malignant (tumor) and non-malignant cells[4–10]. However, scRNA-seq has severe limitations, including technical challenges such as drop-out[11,12] and high cost, which hinder its application to large series and translation to clinical applications.

Several computational deconvolution methods have been developed to predict cellular composition from bulk RNA-seq data by employing a signature of pre-determined cell type-specific gene expression profiles. Initially, these signatures were constructed by sorting each cell type followed by gene expression profiling[13], whereas recent methods such as CIBERSORTx[14] and MuSiC[15] employed scRNA-seq data for this purpose. Most approaches perform linear regression to reconstruct the bulk gene expression profiles using the gene expression signatures, where the regression coefficients correspond to the cellular composition. However, the standard regression approach does not account for variability in gene expression within the same cell type and may render biased results.

To the best of our knowledge, no deconvolution method can adequately and efficiently account for the gene expression variability within the same cell type. Modeling gene expression variability is challenging specifically for deconvolution due to the incompatibility of the log-normalization[16], which significantly stabilizes gene expression variability. Without the log-normalization (i.e., in linear-scale), gene expression data has a heavily skewed distribution, which is not adequately modeled by the standard linear regression approaches, such as non-negative least square (NNLS) used in EPIC[17]. Currently, few probabilistic deconvolution approaches take skewed variability into account. However, these methods handle only a restricted number of cell types due to optimization difficulties (e.g., three cell types in DeClust[18] and Demix/DemixT[19,20]).

Recently, several linear-regression deconvolution approaches have been introduced that consider gene expression variability. MuSiC is a variant of NNLS that prioritizes genes for deconvolution by their variability obtained from the multi-subject single-cell RNA-seq data. CIBERSORTx introduced a two-step approach to address variable cell-type-specific gene expression profiles across the samples: first estimate cellular fraction (deconvolution) and then reconstruct gene expression per cell type in each sample (purification). However, the purification step of CIBERSORTx can handle only a part of genes because of the underdetermination problem where too many parameters need to be inferred. In terms of cellular fraction estimation, both MuSiC and CIBERSORTx outperformed the standard linear regression methods, though they are also linear regression approaches.

Here, we introduce BLADE (Bayesian Log-normAl DEconvolution), a Bayesian method that jointly performs deconvolution and purification in a single-step, taking into account prior knowledge of cell type-specific gene expression profiles obtained from scRNA-seq data. BLADE takes a Bayesian framework that integrates two signatures of mean and variability of gene expression per-cell type using a log-normal probability model. The unified probabilistic model for both deconvolution and purification of BLADE can leverage the prior knowledge for purification, which can remedy the underdetermination issue. Furthermore, an efficient variational inference algorithm was developed, for which we show that it can handle at least 20 cell types. Through a comprehensive evaluation based on more than 700 simulated and real bulk gene expression data sets, we demonstrate a robust performance of BLADE regardless of gene expression variability. In particular, BLADE achieves high accuracy and completeness in gene expression purification, underpinning the power of the unified Bayesian framework for both tasks.

## Results

**Gene expression variability within a cell type.** We first assessed gene expression variability within a cell type using publicly available Peripheral Blood Mononuclear Cell (PBMC) CITE-seq (Cellular Indexing of Transcriptomes and Epitopes by Sequencing) data from 10x Genomics. Based on the integration and clustering analysis followed by phenotyping of 9439 cells, we identified fifteen immune cell types, among which nine are in common, with distinct cell-surface markers and gene expression profiles (Fig. 1a; see "Methods" and Supplementary Figs. S1–2). The size of cell populations ranges from 38 regulatory T cells (0.36%) to 2518 classical monocytes (24%). We then identified differentially expressed genes (DEGs) for each cell type. Subsequently, the standard deviation of gene expression levels per gene and per cell type was measured to assess gene expression variability among the same cell types. We identified high gene expression variability among the same cell populations, especially for DEGs without log-transformation (i.e., linear-scale; Fig. 1b, c). The variability further increased when cells from the two scRNA-seq datasets were combined, indicating the presence of more variability between individuals (Fig. 1d; $P < 2.2 \times 10e-16$ from a one-tailed paired $t$-test of within-sample and between-sample variability).

**Modeling gene-expression variability by probabilistic distribution.** To properly account for variation in gene expression, we examined multiple probability distributions. We evaluated normal distribution, negative binomial distribution, and log-normal distribution to fit the expression level of each gene per cell type without log-normalization. The normal distribution is the standard variability model in many deconvolution algorithms, including CIBERSORTx[14], EPIC[17], and ABIS[21], while the negative binomial distribution is frequently used for handling count data such as RNA-seq data[22]. Note that Poisson distribution was also introduced for modeling count data[23,24], but it is a special case of negative binomial. The log-normal distribution is identical to the normal distribution but includes an exponential function, assuming gene expression data is normally distributed on a log scale but not on a linear scale. To evaluate the performance of these probability distributions on gene expression variability, we assessed (1) the maximum likelihood of fitting gene expression profiles and (2) the difference between estimated and empirical modes (i.e., the most probable gene expression level; Fig. 2a–c). The log-normal distribution, in general, shows the best performance in per-gene maximum likelihood, followed by the negative binomial and normal distributions (Fig. 2a, c). In particular, we noted a biased fit of the normal distribution toward outlier observations, which led to low accuracy in identifying modes (Fig. 2b; see four example genes with a biased fit with normal distribution in Fig. 2d). In mode estimation, log-normal and negative binomial appears to be fairly competitive, where the log-normal had a somewhat worse median but a better third quartile (Fig. 2b).

We further evaluated the performance of the log-normal and negative binomial distributions in the context of deconvolution. To this end, we constructed a generic statistical deconvolution

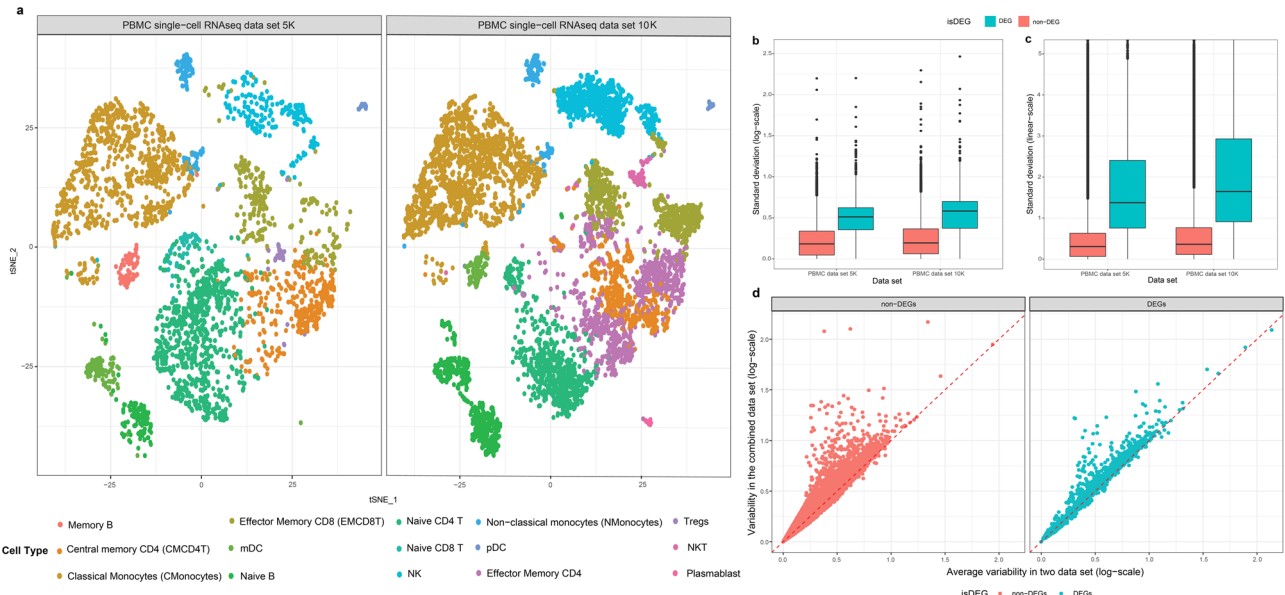

**Fig. 1 Overview of single-cell CITE-seq data from two PBMC samples. a** t-SNE plots show the similarities in Pearson correlation coefficients among gene expression profiles of individual cells in two single-cell PBMC RNA-seq data, respectively, on the left and right. Cell type* is denoted by color. **b, c** Comparison of gene expression variability measured in standard deviation (y-axis) per gene and cell type pair in log-scale (**b**) and linear-scale (**c**) for both datasets (x-axis). The genes were split by differentially expressed genes (DEGs; $n = 2876$ gene and cell type pairs; red) and non-differentially expressed genes (non-DEGs; $n = 145,305$ gene and cell type pairs; blue). The standard boxplot notation was used (lower/upper hinges—first/third quartiles; whiskers extend from the hinges to the largest/lowest values no further than 1.5 * inter-quartile ranges). **d**. Comparison of within-sample (x-axis) and between-sample variability (y-axis) in gene expression levels per cell type, split by DEGs ($n = 2876$) and non-DEGs ($n = 145,305$) per cell type. Standard deviation is measured for each gene and cell type first separately in two PBMC single-cell datasets followed by taking the average (x-axis), then also in merged PBMC data set (y-axis). Only the nine cell types commonly detected in two data sets were used in the analysis. *(CMCD4T: central memory CD4+ T cell; CMonocytes: classical monocytes; EMCD4T: effector memory CD4+ T cell; mDC: myeloid dendritic cell; MemoryB: memory B cell; MemoryCD8T: memory CD8+ T cell; NaiveB: naive B cell; NaiveCD4T: naive CD4+ T cell; NaiveCD8T: naive CD8+ T cell; NKcells: natural killer cell; NKT: natural killer T cell; Nmonocyte: non-classical monocyte; pDC: plasmacytoid dendritic cell; TRegs: regulatory T cell).

method that can model gene expression profiles with various probabilistic assumptions given known cellular fractions. The method approximates the convolution of random variables with an arbitrary distribution using a probabilistic generating function, for which both negative binomial and log-normal random variables can be accurately approximated (see "Methods", Supplementary Note 1, and Supplementary Fig. S3). Based on this method, we evaluated the performance of negative binomial and log-normal distribution in fitting the gene expression profiles per cell type using RNA-seq data from TCGA[25]. First, we obtained TCGA RNA-seq data of mesothelioma (TCGA-MESO; $n = 84$) and sarcoma (TCGA-SARC; $n = 256$), from which we estimated the fraction of eight cell types using EPIC[17], a deconvolution method previously applied to the TCGA. Then, we applied the flexible deconvolution method with two different probabilistic assumptions, log-normal and negative binomial, to estimate expression profiles per cell type of 200 random genes. In terms of log-likelihood and root mean square error (RMSE) measured per gene, log-normal and negative binomial deconvolutions performed equally well for most of the genes, except for a few genes (Fig. 2e, f). Cumulatively, we concluded that the log-normal distribution is an attractive probabilistic distribution to model the gene expression variability of each cell type.

**Overview of BLADE: Bayesian Log-normal Deconvolution.** We constructed a Bayesian Log-normal Deconvolution method, BLADE, by emulating bulk gene expression profiles through convolution of gene expression profiles per cell type (Fig. 3a). The bulk gene expression level of each gene $j$ in sample $i$ was modeled by $y_{ij} = \sum_t f_i^t x_{ij}^t + \epsilon_{ij}$. Here, the hidden variables $f_i^t$ and $x_{ij}^t$

denote the fraction of cell type $t$ for sample $i$ and the purified expression level of gene $j$ of cell type $t$ for sample $i$. These hidden variables $f_i^t$ and $x_{ij}^t$ are, respectively, endowed with the Dirichlet distribution and the log-normal distribution. To incorporate prior knowledge from scRNA-seq data, we take a hierarchical approach to model $x_{ij}^t$ by taking a conjugate prior of log-normal distribution with hyperparameters $\mu_{0j}^t$, $\kappa_{0j}^t$, $\alpha_{0j}^t$, and $\beta_{0j}^t$ (Fig. 3b). The hyperparameters are chosen based on the mean and standard deviation of each gene per cell type from the scRNA-seq data. By inferring the hidden variables, we can jointly estimate the fraction of cell types, captured by $f_i^t$, and purified gene expression profiles of each cell type in each sample, captured by $x_{ij}^t$. For inference, we employed a collapsed variational inference that maximize efficiency by integrating out a subset of hidden variables with a conjugate prior in advance. Furthermore, we employed the L-BFGS algorithm in conjunction with machine-code translated Python code for gradient and objective function calculations instead of native Python code. The compilation of native Python code by the Numba package[26] significantly accelerates gradient and objective functions that are executed thousands of times during the L-BFGS optimization (Supplementary Fig. S4). See "Methods" and Supplementary Note 2 for further details of the framework. As a result, BLADE can handle many cell types (>20 cell types); unlike the previous log-normal-based deconvolution that can account for a maximum of three cell types[20].

**Robustness of BLADE deconvolution against gene expression variability.** We assessed the robustness of BLADE, CIBERSORTx, and non-negative least squares (NNLS) against gene expression

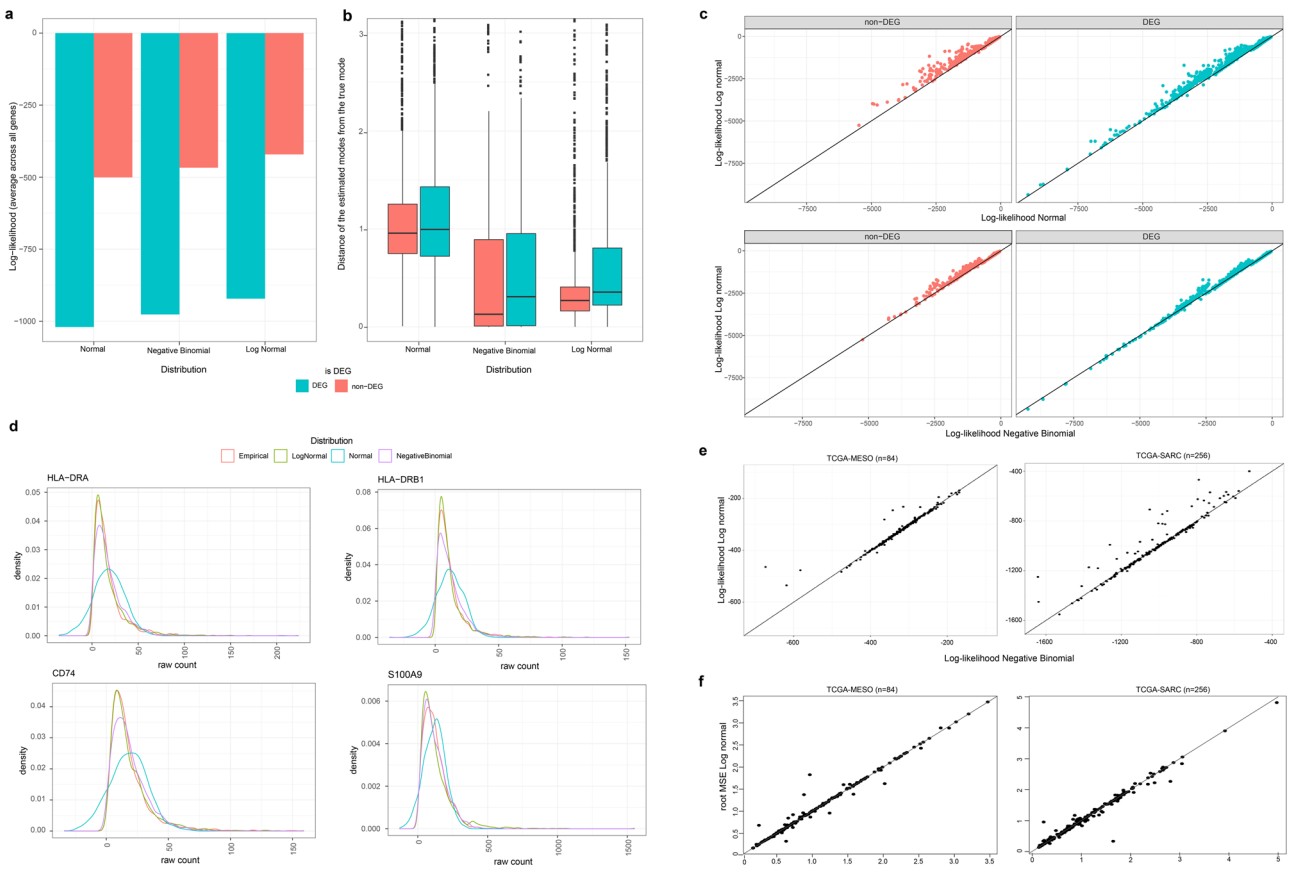

**Fig. 2 Comparison of normal, negative binomial, and log-normal distribution in fitting linear-scale gene expression data. a** A bar chart of average log-likelihood of the three types of distribution fitted to PBMC single-cell RNA-seq data. The genes were split by DEGs (red; $n = 1723$) and non-DEGs (blue; $n = 1496$). **b** Comparison of the distance of the estimated mode to the true mode (y-axis) per distribution type (x-axis). The standard boxplot notation was used (lower/upper hinges— first/third quartiles; whiskers extend from the hinges to the largest/lowest values no further than 1.5 * inter-quartile ranges). **c** Pairwise comparison of per-gene log-likelihood of log-normal distribution (y-axis) and that of normal (x-axis; top) and negative binomial distribution (x-axis; bottom). The genes were split into non-DEGs (left) and DEGs (right). **d** Density plots for raw-counts (red) and optimized log-normal (green), normal (blue), and negative binomial distribution (purple) for four example genes (gene name at the top) with low maximum log-likelihood for normal distribution. **e, f** Maximum log-likelihood values (**e**) and root mean squared error (root MSE: **f**) of each gene for log-normal (y-axis) and negative binomial (x-axis) convolutions of $T = 8$ cell types, applied to TCGA-MESO (left) and TCGA-SARC (right) data.

variability by applying them to model-based simulation data. The simulation data was created to have diverse but controlled variability levels of gene expression profiles (standard deviation of 0.1–1.5) as well as different numbers of cell types (5–20 cell types), marker genes (100–1000 genes), and samples (5–100 samples; in total 700 training data sets). Note that NNLS is a regularized linear regression, a type of constrained linear regression used in many deconvolution methods, including MuSiC[15], EPIC[17], TIMER[27], ABIS[21], and also in the purification step of CIBERSORTx[14]. The simulation data variability levels were selected to recapitulate the observed range in the scRNA-seq data (up to standard deviation of 1.5 in log scale; Fig. 1b, c). In general, all three methods could accurately estimate cellular fractions in case of a high number of genes, a low number of cell types, and a low variability level. In contrast, the performance decreased when a smaller number of genes are presented, and the number of cell types is increased (Fig. 4a; Supplementary Figs. S5–7). However, BLADE was the most robust against gene expression variability. In particular, in the range of observed expression variability of DEGs in the PBMC scRNA-seq data (on average standard deviation of > 0.5; Fig. 1b), BLADE significantly outperformed CIBERSORTx and NNLS.

We then compared the performance of BLADE and CIBER-SORTx in estimating gene expression profiles per cell type. In this comparison, NNLS is not included because of redundancy since the purification step of CIBERSORTx is based on NNLS. There are two modes of purification in CIBERSORTx, both of which were compared with BLADE: (1) estimating average profile per cell type across the samples (group mode purification), and (2) estimating the profile per cell type for each sample (high-resolution mode purification). For the data set with low variability levels, both BLADE and CIBERSORTx accurately reconstructed gene expression profiles per cell type (Fig. 4b, c; Supplementary Figs. S8–9). However, unlike BLADE, the performance of CIBERSORTx decreased rapidly as the RNA expression variability within a cell type increased. Furthermore, CIBERSORTx often excludes genes for purification, especially in high-resolution mode, when: (1) the number of cell types is larger than or equal to the number of samples, and (2) the variability in gene expression is high (Fig. 4d; Supplementary Figs. S10, S11). BLADE could accurately estimate the gene expression profiles of each cell type in both group mode and high-resolution mode, regardless of the number of cell types and samples, without any filtering (Fig. 4b, c; Supplementary Figs. S8–9).

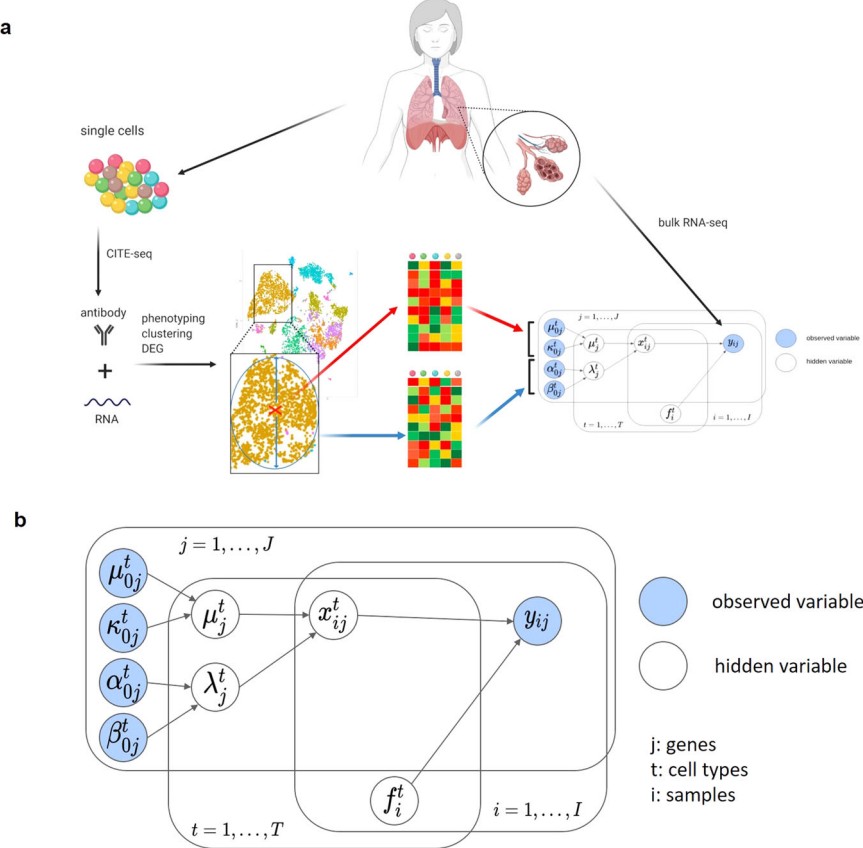

**Fig. 3 BLADE workflow. a** To construct a prior knowledge for BLADE, we used CITE-seq data that contains gene expression and cell surface marker profiles of each cell. Cells are then subject to phenotyping, clustering, and differential gene expression analysis. Then, for each cell type, we retrieved average expression profiles (red cross and top heat map) and standard deviation per gene as the variability (blue circle and bottom heatmap). This prior knowledge is then used in the hierarchical Bayesian model (bottom right) to deconvolute bulk transcriptome profiles. **b** A graphical model of BLADE represents random variables, observed and hidden variables, respectively, in blue and gray nodes, and their dependency associations (arrows). See the text for the details of the model.

**Application of BLADE to in silico mixture of PBMC scRNA-seq data**. We constructed realistic bulk gene expression data by in silico mixing the scRNA-seq data from PBMC samples without any model assumption to further evaluate our method. To this end, we randomly sample 100 cells 20 times from the 9439 cells from the two PBMC scRNA-seq data sets. We chose to use 100 cells since more cells get selected commonly in multiple samples as we sample more, making the simulated bulk gene expression data lose variability between the samples. In order to make the simulation data as realistic as possible, a cumulative sum of raw counts of 100 cells was obtained, followed by a standard normalization. The resulting simulation data recapitulate the gene expression variability of 15 cell types (Fig. 5a; Supplementary Fig. S12). We constructed signature matrices that capture the true mean and the standard deviation of 1007 genes selected and measured using all of 9439 cells (top 200 DEGs with FDR < 0.2 per cell type, combined). We also generated three extra data sets with a coarse classification of the 15 cell types by four (level 1; 441 genes selected), eight (level 2; 604 genes), and 12 cell types (level 3; 880 genes) in the same manner to diversify the difficulty levels for deconvolution (see Supplementary Data 1 for the details of classifications). The increase of cell type often lowers the fraction of each cell type and the number of genes that can classify each cell type (Supplementary Figs. S13–14). In particular, the fraction of T cells in level 1 is 0.47 on average, which gets much lower for their subtypes in level 4 (0.01 and 0.094 on average for naive and

memory CD8+ T cells; Supplementary Fig. S13). Furthermore, although more genes selected in the higher levels, there are 25 unique DEGs for T cells (DEGs only identified for T cells) in level 1, whereas there are only 16 and 3 unique DEGs for naive and memory CD8+ T cells in level 4 (Supplementary Fig. S14). Collectively, deconvolution gets more challenging as the number of cell types increases from level 1 to level 4.

Using the bulk PBMC data generated above, we evaluated BLADE taking CIBERSORTx, NNLS, and also MuSiC as the baseline. We used the same list of genes and signatures for the baseline methods for a fair comparison. In general, the accuracy of estimated cell type fractions gets lower as the number of cell types gets higher, as expected (Fig. 5b, see also Spearman correlation coefficients and RMSE in Supplementary Fig. S15). All algorithms reached > 0.5 Pearson correlation coefficient for almost all cell types at level 1, where many cell types failed to reach as high performance as the number of cell types increased. Interestingly, the performance was sometimes higher in level 3 than level 2, especially for MuSiC, possibly because the advantage of having more genes overcomes the complexity due to the increased number of cell types (e.g., 880 genes in level 3, compared to 604 genes in level 2). At level 4, BLADE outperformed CIBERSORTx (P-value of 0.0087; a one-tailed paired t-test) and NNLS (P-value of 0.021; a one-tailed paired t-test) and performed comparably to MuSiC (P-value of 0.46; one-tailed paired t-test). The performance of the four methods are significantly correlated (P-value < 0.05 from Pearson correlation

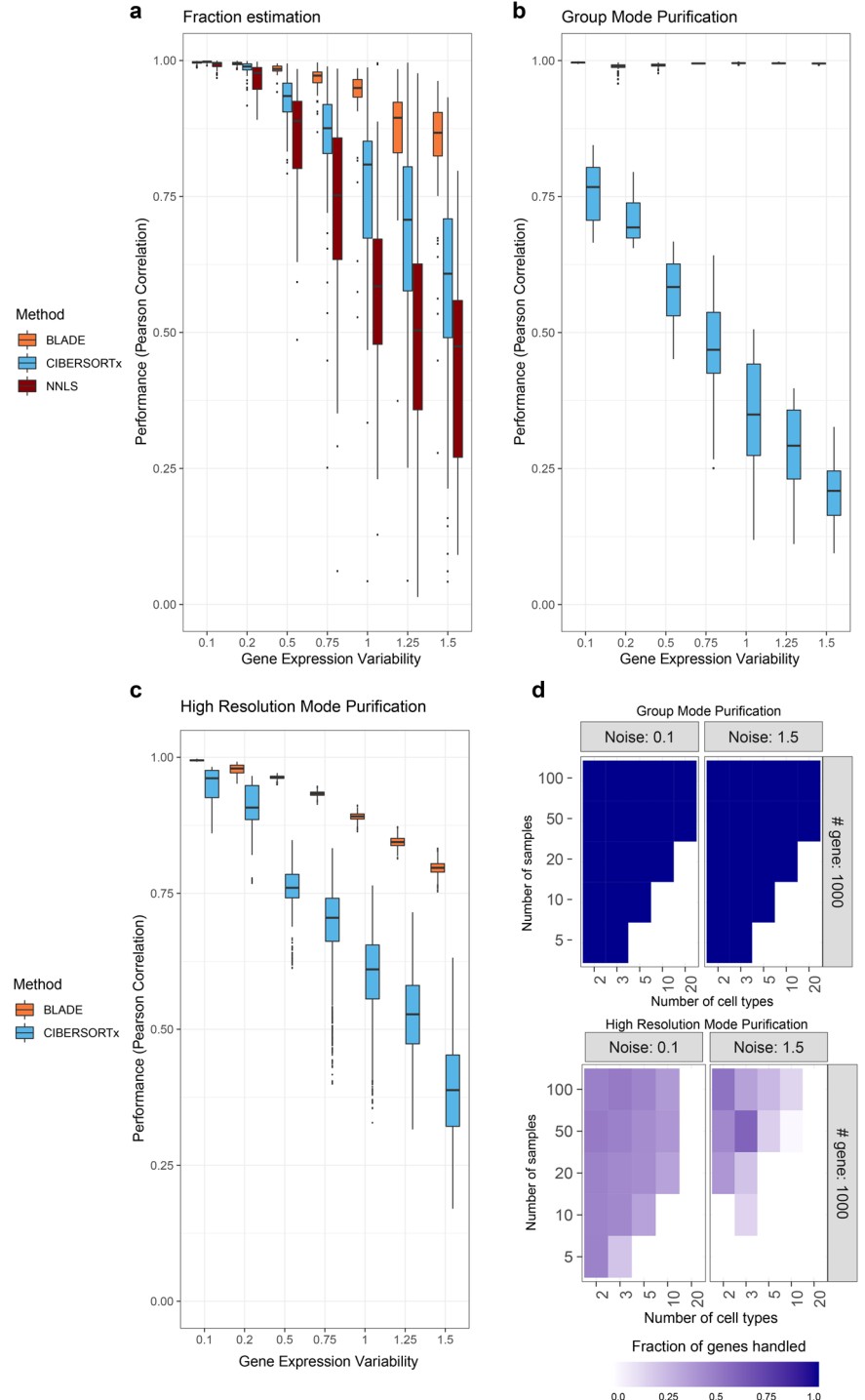

**Fig. 4 Performance evaluation BLADE using simulation data with diverse settings. a** Performances (Pearson correlation coefficient; *y*-axis) of BLADE (orange), CIBERSORTx (blue), and NNLS (dark red) to predict the cellular fraction of a subset of simulation data with ten cell types, 1000 genes, and various variability levels (standard deviation of 0.1–1.5; *x*-axis; *n* = 50 per variability level; five independent data set with ten cell types each). The standard boxplot notation was used (lower/upper hinges—first/third quartiles; whiskers extend from the hinges to the largest/lowest values no further than 1.5 * inter-quartile ranges). **b**, **c** Performances (Pearson correlation coefficient; *y*-axis) of BLADE (orange) and CIBERSORTx (blue) to predict gene expression profiles per cell type for all samples jointly (group mode; **b**) and for each sample separately (high-resolution mode; **c**) using the same simulation data (*n* = 50 per variability level; five independent data set with ten cell types each). The standard boxplot notation was used. **d** Fractions of purified genes in the simulation data with two extreme levels of gene expression variability (left and right panels) by CIBERSORTx in group mode (top) and high-resolution mode (bottom). *x*- and *y*-axis represent the number of cell types and samples in the simulation data, respectively.

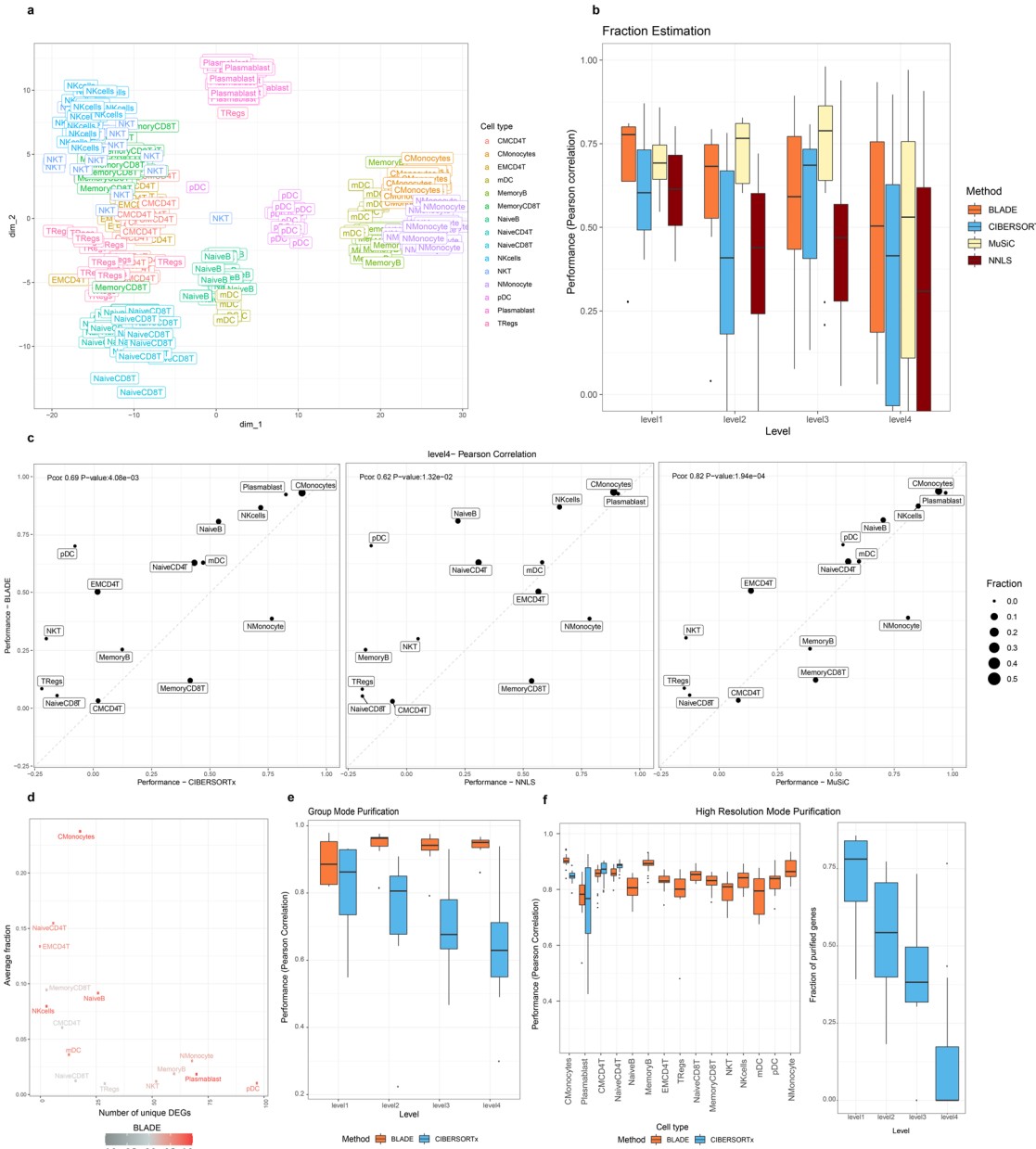

**Fig. 5 Performance evaluation of BLADE using simulated PBMC bulk RNA-seq data. a** A t-SNE plot represents the similarities in Pearson correlation coefficients among gene expression profiles of 15 cell types* (denoted by label) in 20 simulated bulk PBMC data. **b** Performances (Pearson correlation coefficient; y-axis) of BLADE (orange), CIBERSORTx (blue), NNLS (dark red), and MuSiC (light yellow) in predicting cellular fractions of the 20 simulated PBMC bulk RNA-seq data with diverse levels ($n = 4$, 7, 12, and 15 cell types, respectively, in levels 1–4; x-axis). The standard boxplot notation was used (lower/upper hinges —first/third quartiles; whiskers extend from the hinges to the largest/lowest values no further than 1.5 * inter-quartile ranges). **c** Comparison of performance in estimating the cellular fractions per cell type of BLADE (y-axis) with CIBERSORTx, NNLS, and MuSiC (x-axis) at level 4. The fraction of each cell type is indicated by the size of the point. Pearson correlation coefficient and two-tailed test P-values are indicated at the top left in each panel. **d** Performance of BLADE (indicated by color) and its association to the number of unique DEGs per cell type (x-axis) and the respective fraction in the simulated data (y-axis). **e** Performance in Pearson correlation coefficient of BLADE (orange), CIBERSORTx (blue) for group mode purification of four levels of PBMC simulation data ($n = 4$, 7, 12, and 15 cell types, respectively, in levels 1–4; x-axis). The standard boxplot notation was used. **f** Performance (Pearson correlation coefficient; y-axis) of BLADE (orange) and CIBERSORTx (blue) in estimating gene expression profiles per cell type (x-axis) and per sample in level 4 ($n = 20$ samples per cell type; left). Fraction of genes in silico purified in high-resolution mode by CIBERSORTx at all levels of PBMC simulation data ($n = 20$ samples with 4, 7, 12, and 15 cell types, respectively, in levels 1–4; x-axis; right). The standard boxplot notation was used. *(CMCD4T: central memory CD4+ T cell; CMonocytes: classical monocytes; EMCD4T: effector memory CD4+ T cell; mDC: myeloid dendritic cell; MemoryB: memory B cell; MemoryCD8T: memory CD8+ T cell; NaiveB: naive B cell; NaiveCD4T: naive CD4+ T cell; NaiveCD8T: naive CD8+ T cell; NKcells: natural killer cell; NKT: natural killer T cell; Nmonocyte: non-classical monocyte; pDC: plasmacytoid dendritic cell; TRegs: regulatory T cell).

test), especially in pairs of MuSiC and BLADE (Pearson correlation coefficient = 0.82; *P*-value = 1.9e−04), and NNLS and CIBERSORTx (Pearson correlation coefficient = 0.87; *P*-value = 3.0e−05; Fig. 5c; Supplementary Fig. S16 for the comparison in the levels 1–3). Among the 15 cell types, plasmablasts, classical monocytes, natural killer (NK) cells were the best predicted by all four methods, which commonly failed to predict the composition of regulatory T cells (Tregs), naive CD8+ T cells (NaiveCD8T), and central memory CD4+ T cells (CMCD4T). These cell types are commonly low abundant (fraction of < 7% on average), and only a few unique DEGs were identified for each cell type (< 50 unique DEGs; Fig. 5d; see Supplementary Fig. S17 for other levels). In contrast, we noted a decent predictive performance of all methods for the abundant cell types (> 10%) with a high number of DEGs (> 50 unique DEGs).

BLADE significantly outperformed CIBERSORTx in estimating gene expression profiles per cell type in both group mode and high-resolution mode across all the levels (Fig. 5e, f and Supplementary Fig. S18). For group mode purification, CIBER-SORTx performed comparably to BLADE at level 1, which, however, gets lower at the higher level. Here, BLADE's performance was near-perfect, as expected, since BLADE integrates cell-type-specific gene expression profiles for purification (Fig. 5e). CIBERSORTx did not estimate expression levels of most genes in high-resolution mode, and essentially no genes were purified for 11 cell types at level 4 (right panel of Fig. 5f; Supplementary Fig. S19). Furthermore, estimated expression profiles by CIBERSORTx are in general less accurate than BLADE in all levels, except for few cell types (e.g., central memory CD4+ T cells and naive CD4+ T cells at level 4; Fig. 5f). The performance of BLADE in high-resolution mode purification is consistently accurate (> 0.7 Pearson correlation coefficient) across all cell types in all levels (Supplementary Fig. S20). Cumulatively, Bayesian simultaneous deconvolution and in silico purification by BLADE significantly outperformed CIBERSORTx in reconstructing gene expression profiles per cell type.

**Application of BLADE to standard bulk RNA-seq data with incomplete prior knowledge.** We further challenged BLADE and other deconvolution algorithms using the standard bulk RNA-seq data of PBMC immune cell mixtures for which the composition of eight immune cell types was determined by flow cytometry[28] (Fig. 6a). Of these eight cell types, neutrophils were not identified in our PBMC scRNA-seq data. Furthermore, there are undetermined cells by the flow cytometry analysis that still contributed to the bulk RNA-seq data. Therefore, there is only limited prior knowledge available on cell-type-specific gene expression profiles, which is the case for most applications of deconvolution. We applied BLADE and other baseline methods using the gene expression signatures consisting of 532 genes that can distinguish seven cell types derived in the same manner as in the previous section (see Supplementary Data 1 for the cell type classification). BLADE was able to reconstruct fractions of the seven cell types rather accurately, except for myeloid dendritic cells (mDC; Fig. 6b, c and Supplementary Fig. S21). We confirmed a low concordance of mDC signature compared to the previously determined signature using a large number of RNA-seq data[28] (53 samples; Fig. 6d). In fact, mDC signature has a higher correlation with previous B cell and monocyte signatures (Fig. 6e), which makes the signatures less informative and the deconvolution extra challenging. Other baseline methods estimated compositions of monocytes accurately, but they failed to do the same for the majority of the other cell types including mDC (Fig. 6b). In fact, they often failed to detect some cell types, particularly

Tregs are commonly missed (Fig. 6c). Instead, BLADE over and underestimated the fractions of Tregs and CD8+ T cells, respectively, by absorbing CD8+ T cell fractions to Tregs. However, BLADE was still able to rank samples accurately by their fractions. Cumulatively, BLADE was the most robust method for estimating cell type fractions when available prior knowledge was incomplete.

**Evaluation of BLADE for deconvolution of tumor RNA-seq data.** We further evaluated our method using scRNA-seq data from tumor samples. First, we obtained scRNA-seq data for 35 pancreas samples (CRP000653; Genome Sequence Archive), of which 24 are tumors while the other 11 are normal. The scRNA-seq data contains 57,530 cells classified into 10 cell types[29] (Fig. 7a; Supplementary Fig. S22). For a fair evaluation of deconvolution algorithms, the 35 samples and their cells were split into auxiliary (six samples, of which four are tumors) and main samples (29 samples, of which 20 are tumors; Supplementary Fig. S23). From the auxiliary samples, we obtained the mean and standard deviation of 818 genes that can classify ten cell types reliably (top 100 DEGs with FDR < 0.1 per cell types). For the main samples, we generated bulk gene expression profiles by calculating a cumulative sum of the raw count of all cells, followed by the standard log-normalization. For predicting the fraction of 10 cell types, MuSiC performed the best, followed by BLADE and CIBERSORTx (Fig. 7b; see Spearman correlation coefficients and RMSE in Supplementary Fig. S24). Interestingly, the performance of BLADE correlates the most with MuSiC (Pearson correlation coefficient of 0.62; *P*-value of 0.056), whereas it is less so with CIBERSORTx (Pearson correlation coefficient of 0.39; *P*-value of 0.27) and NNLS (Pearson correlation coefficient of −0.18; *P*-value of 0.62; Fig. 7c). BLADE outperformed MuSiC for predicting the fraction of B cells but was worse for predicting endocrine cell fractions. Most cell types achieved high performance (> 0.5 of Pearson correlation coefficient) in all methods, except for B cells (in MuSiC and CIBERSORTx), T cells (in CIBERSORTx and NNLS), and Stellate cells (in NNLS). These cell types are often less dominant (< 5%) and with a small number of DEGs (less than 40 unique DEGs; Fig. 7d). For reconstructing cell-type-specific gene expression profiles, both BLADE and CIBERSORTx achieved high performance for all cell types (> 0.8 of Pearson correlation coefficient; > 0.9 mostly for BLADE; Fig. 7e, f). However, the purification by BLADE is without any filtering, unlike CIBERSORTx, which purified around 30% of genes per cell type on average in high-resolution mode (Fig. 7g). Cumulatively, BLADE is a reliable deconvolution method especially to reconstruct cell-type-specific gene expression profiles in the tumor context.

**Discussion**
One of the major challenges in the deconvolution of bulk RNAseq data is the adequate and efficient handling of gene expression variability, mainly since stabilization of variability by log-normalization is inapplicable. Most of the previous algorithms implicitly or explicitly assumed normal distribution, as otherwise, the inference is highly challenging and limits the number of cell types that can be handled maximally (three cell types in Demix[19]). However, the normal distribution often renders a biased fit for gene expression variability (Fig. 2a–d), leading to a suboptimal outcome of deconvolution algorithms. Consequently, the performance of the standard regression technique, NNLS, was consistently inferior, especially when there is a realistic level of gene expression variability (Figs. 4–7).

CIBERSORTx and MuSiC are also linear-regression approaches that partially alleviate the issue by prioritizing genes for

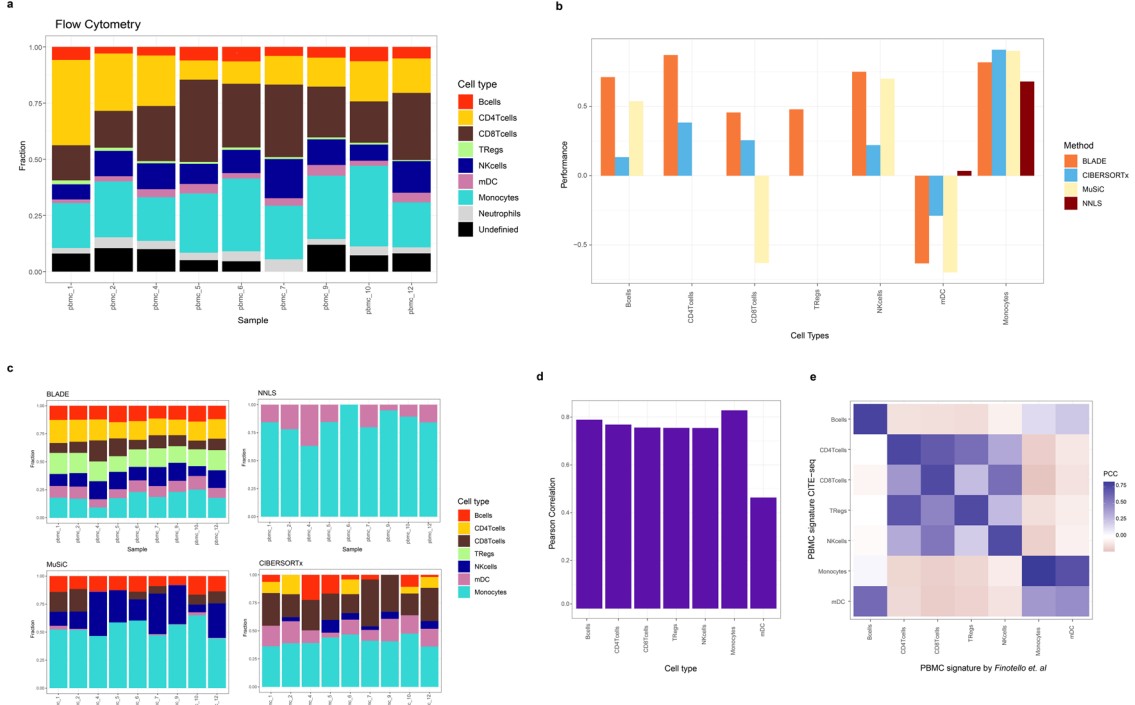

**Fig. 6 Performance evaluation of BLADE using PBMC bulk RNA-seq data with incomplete prior knowledge. a** Cell types fractions (y-axis) determined by flow cytometry in nine samples (x-axis). All cell types* have a color associated as shown in the legend. **b** Performances of BLADE (orange), CIBERSORTx (blue), NNLS (dark red) and MuSiC (light yellow), measured by Pearson correlation (y-axis) of the estimated sample-specific cell type (x-axis) fractions with those determined by flow cytometry. **c** Estimated cell fractions (y-axis) per sample (x-axis) by BLADE (top-left), NNLS (top-right), MuSiC (bottom-left) and CIBERSORTx (bottom-right). **d, e** Pearson correlation (y-axis in **d** and color gradient in **e**) of the signature per pair of cell types determined by Finotello et al. and two PBMC scRNA-seq data used in this study. *(TRegs: regulatory T cells; NKcells: natural killer cells; mDC: myeloid dendritic cells).

deconvolution. Support vector regression, the core algorithm of CIBERSORTx, depends on a subset of genes with high reconstruction errors. On the contrary, MuSiC explicitly learns gene weights from the single-cell RNA-seq data and prioritizes genes with low variability, for which the normal distribution can fit accurately due to the low skewness. We noted a consistently superior performance for fraction estimations of these algorithms over NNLS (Figs. 4a, 5b, 6b, 7b). MuSiC outperformed BLADE in some cases, indicating normal distribution-based deconvolution can also be accurate when genes are prioritized based on the gene expression variability. However, performance of MuSiC compared unfavorably to that of BLADE when prior knowledge was incomplete (Fig. 6). Furthermore, the strategy of prioritizing genes reduces the completeness of the purification results (Figs. 4d, 5f, 7g). We observed a lower performance of linear regression-based purification by CIBERSORTx, particularly in high-resolution mode, which may be due to the inefficient variability model and a large number of variables to be estimated (Figs. 4b, c, 5e, f, 7e, f).

BLADE is a hierarchical Bayesian model that simultaneously performs deconvolution and estimation of gene expression profiles per cell type. The log-normal convolution model efficiently accounts for variability in gene expression and also for prior knowledge of gene expression profiles per cell type derived from scRNA-seq data (Fig. 3). Notably, thanks to the unified probabilistic model used in BLADE, the prior knowledge contributes to both deconvolution and gene expression purification. This prior knowledge significantly reduces the search space of solutions for both tasks, which leads to enhanced accuracy and completeness, especially for gene expression purification. The efficient variational inference of BLADE allowed it to handle many cell types while accurately modeling the gene expression

variability. Furthermore, the hierarchical approach of BLADE makes it robust against the quality of prior knowledge, as demonstrated in Fig. 6. Finally, unlike MuSiC and CIBERSORTx, the Bayesian framework of BLADE provide the uncertainties of estimates, which may be valuable to evaluate the quality of the results and for further downstream analysis.

Enhanced in silico microdissection by BLADE opens up the possibility to molecularly characterize individual cell types in tissue based on the standard RNA-seq data. For instance, we demonstrated that BLADE could be applied to estimate each cell type's gene expression profiles that make up the tumor microenvironment (TME). This allows us to characterize pathway activity in each immune cell type and possibly to recognize additional cell (sub-)types. Furthermore, BLADE can aid previously established gene expression subtypes (e.g., PDAC[30,31]) by characterizing the subtypes with distinct TME profiles. Finally, the detailed profiling of the TME, particularly immune TME profiles, may lead to a clinically applicable biomarker strategy for immunotherapy based on the standard bulk gene expression profiling. In conclusion, BLADE is a powerful tool that can significantly contribute to unravel cellular heterogeneity in complex biological systems.

## Methods

**PBMC single-cell RNA-seq data**. Two public peripheral blood mononuclear cell (PBMC) CITE-seq (cellular indexing of transcriptomes and epitopes by sequencing) datasets of healthy donors were downloaded from 10x Genomics datasets database [https://support.10xgenomics.com/single-cell-gene-expression/datasets/3.0.2/5k_pbmc_protein_v3] [https://support.10xgenomics.com/single-cell-gene-expression/datasets/3.0.0/pbmc_10k_protein_v3]. Genes and cells were filtered based on the following criterions: percentage of mitochondrial genes <10% and number of genes per cell between 200 and 4000. After the filtering, raw count data was normalized and scaled, using SCTransform, which performs normalization and variance stabilization using regularized negative binomial regression.

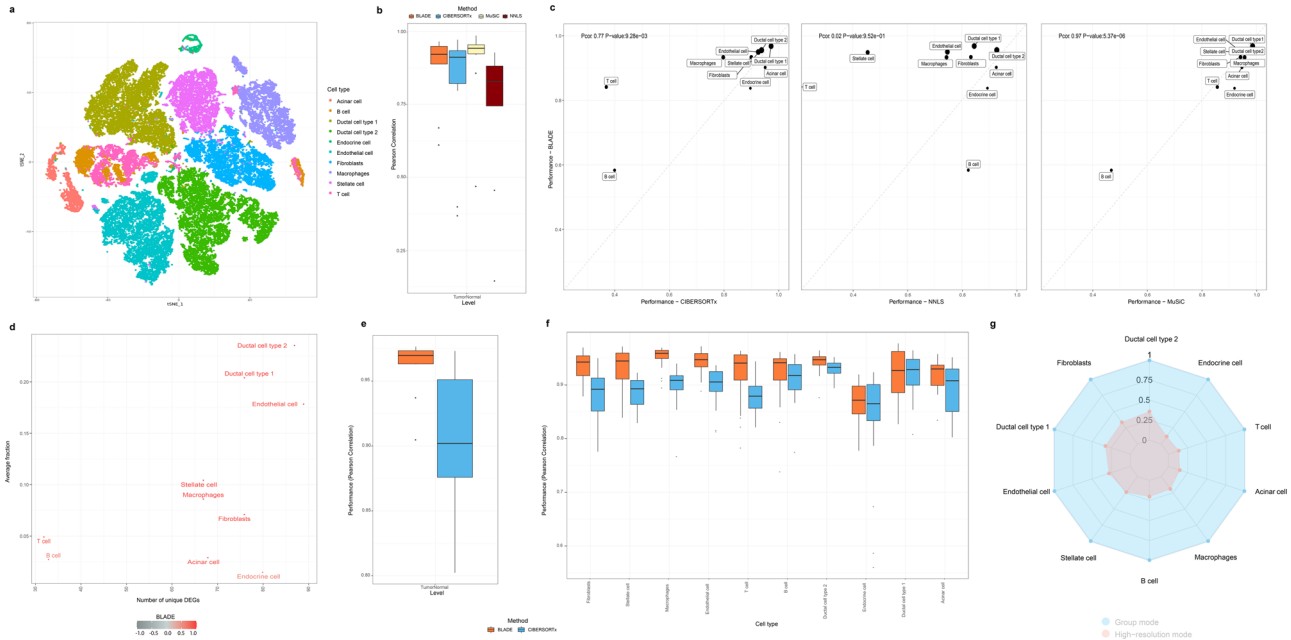

**Fig. 7 Performance evaluation of BLADE for deconvoluting tumor data. a** t-SNE plot showing the variability of the cell populations in the PDAC single-cell RNA-seq data. **b** Performances (Pearson correlation coefficient; y-axis) of BLADE (orange), CIBERSORTx (blue), NNLS (dark red), and MuSiC (light yellow) in predicting cellular fractions of the PDAC bulk RNA-seq data ($n = 29$ main samples). The standard boxplot notation was used (lower/upper hinges— first/third quartiles; whiskers extend from the hinges to the largest/lowest values no further than 1.5 * inter-quartile ranges). **c** Comparison of performance in estimating the cellular fractions of each cell type of BLADE (y-axis) with CIBERSORTx (left), NNLS (middle), and MuSiC (right; x-axis). The fraction of each cell type is indicated by the point size. Pearson correlation coefficient and two-tailed test P-values are indicated at the top left in each panel. **d** The number of unique DEGs per cell type (x-axis) and the respective fraction in the PDAC data (y-axis). **e** Performance in Pearson correlation of BLADE (orange), CIBERSORTx (blue) for group mode purification ($n = 10$ cell types). The standard boxplot notation was used. **f** Performance (Pearson correlation coefficient; y-axis) of BLADE (orange) and CIBERSORTx (blue) in estimating gene expression profiles per cell type (x-axis) and per sample ($n = 29$ samples for each cell type; right). The standard boxplot notation was used. **g** Fraction of genes in silico purified in group mode (blue) and high-resolution mode (blue) by CIBERSORTx.

Dimensionality reduction was done using principal component analysis (PCA) and t-distributed stochastic neighbor embedding (t-SNE). Following that, k-nearest neighbors (knn) of each cell using 25 dimensions of PCA were determined. This knn graph was used to construct the Shared Nearest Neighbor (SNN) graph by calculating the neighborhood overlap (Jaccard index) between every cell and its 20 nearest neighbors. Cluster determination was done by SNN graph modularity optimization based on the Louvain algorithm with the resolution of 1. Cells were phenotyped separately in both datasets, using primarily cell surface markers and then gene expression levels in case of lack of usable cell surface markers (Supplementary Figs. S1–2). The two datasets were individually normalized, followed by selecting variable genes. The two data set were then integrated, and batch corrected using the common variable genes. The same analysis as described above was performed on the merged data set, including PCA, SNN, and cluster determination[32]. Finally, the top 200 differentially expressed genes per cell type were identified using a two-sided Wilcoxon Rank sum test by taking a contrast between one cell type versus the rest with an FDR cutoff of 0.2.

**Comparison between log-normal, negative binomial, and normal distribution in fitting raw gene expression counts.** To evaluate log-normal, normal, and negative binomial distribution in fitting gene expression profiles, we retrieved raw counts per gene and per cell type and fit the three distribution types using the maximum-likelihood method available in *fitdistrplus* R package. For each cell type, genes with a standard deviation of lower than 0.5 were filtered out as they are mostly not expressed in that cell type. Finally, the log-likelihoods of the optimized distributions were obtained per gene and per cell type for comparison. As an alternative measure, we also identified the mode (i.e., the peak of the probability distribution) in each of the optimized distributions and assessed its accuracy by comparing it to the mode of the empirical distribution for each gene and cell type pair.

**A generic deconvolution method with known cellular composition.** For a fair comparison of log-normal and negative binomial distribution for deconvolution, we developed a simple, generic maximum-likelihood-based convolution model. Formally it is assumed that there are $i = 1, ..., I$ samples in which $t = 1, ..., T$ cell types jointly contribute to expression profiles of $j = 1, ..., J$ genes. For each sample $i$

and gene $j$, a bulk expression level is given, indicated by $y_{ij}$. Then, two hidden variables were introduced that jointly makeup $y_{ij}$: (1) expression level of the gene per cell type and sample, $x^t_{ij}$; and (2) cellular composition for each cell type $t$, $f^t_i$, where $\forall f^t_i \geq 0$ and $\sum_t f^t_i = 1$. An important strength of our method here is that it applies to any underlying parametric distribution for $x^t_{ij}$. $y_{ij}$ is a (weighted) convolution:

$$y_{ij} = \sum_{t=1}^{T} f^t_i x^t_{ij} \quad (1)$$

which implies, with $\hat{x}^t_{ij} = f^t_i x^t_{ij}$,

$$g_{y_{ij}}(y) = \int_{u_1=0}^{y} \cdots \int_{u_T=0}^{y-\sum_{i=1}^{T-1} u_i} g_{\hat{x}^1_{ij}}(u_1) \cdots g_{\hat{x}^{T-1}_{ij}}(u_{T-1}) g_{\hat{x}^T_{ij}}\left(y - \sum_{t=1}^{T-1} u_t\right) du_1 \cdots du_T. \quad (2)$$

By assuming $x^t_{ij}$ follows log-normal distribution (i.e., $x^t_{ij} \sim LN(\mu^t_j, (\sigma^t_j)^2)$) and thus $\hat{x}^t_{ij} \sim LN(\mu^t_j + \log f^t_i, (\sigma^t_j)^2)$, $y_{ij}$ is a convolution of $T$ log-normal random variables. The interest lies in estimating parameters $\theta_j = (\mu^t_j, \sigma^t_j)$ by maximum likelihood.

While numerical evaluation of (2) may still be efficient for $T = 2$[28], however, the extension to $T > 2$ is not straightforward to a $T − 1$ dimensional integral. To this end, the log-normal density $g_t = g_{\hat{x}^t_{ij}}$ is approximated by a probability generating function (PGF). See Supplementary Note 1 for the details of PGF approximation. The PGF-based approximation of $g_t$ showed higher accuracy than an alternative approximation method, Fenton-Wilkinson (FW) approximation[33], which was also included as a benchmark (see Supplementary Note 1 and Supplementary Fig. S3).

**Comparison of LN and NB based on the generic deconvolution technique.** The aforementioned generic deconvolution was used to evaluate LN and NB for deconvolution. For this, two RNA-seq data sets are retrieved from The Cancer Genome Atlas (https://tcga-data.nci.nih.gov/tcga/) using TCGAbiolinks[34]. We considered all complete samples from the following tumor types: Mesothelioma (MESO[35], $n = 84$; and Sarcoma (SARC[36], $n = 256$. We retrieved the upper quartile

normalized RSEM (RNASeq by expectation-maximization) TPM (transcript per million) gene expression values (R package curatedTCGAData), merged replicated measurements (R package MultiAssayExperiment), and extracted the sample definitions from the barcodes (R package TCGAutils). We retained genes with mean count larger or equal to 5. For visualizing results, 200 genes were sampled randomly from this set[37]. The comparison procedure for LN and NB distributions is:

1. Apply a non-statistical method, EPIC[17], to estimate cell type fractions for bulk RNA-seq data using cell type-specific reference signatures. It has shown that EPIC provides a reliable estimate of cellular fractions of $T = 8$ cell types[38], and it provides fractions that add up to 1.

2. Fix the cellular fractions and fit generic deconvolution models with $T = 8$ LN or NB components using maximum likelihood.

3. Compare the maximum likelihood values of the LN and NB models for of $J$ genes.

The above procedure was done for 200 randomly selected genes with mean count per million larger or equal to 5 to exclude lowly expressed genes. Note that the comparison of the maximum likelihood values is fair, because the number of parameters used in the LN and NB components is the same, of $2T = 16$ per gene. As an alternative metric, we also measured the accuracy in reconstructing bulk gene expression levels based on deconvolution. Taking actual and predicted bulk gene expression level in LN or NB deconvolution model, root-mean-squared error (RMSE) was evaluated per gene and per model.

**Hierarchical Bayesian model for convolution of log-normal variables (BLADE).** BLADE is a hierarchical Bayesian model for log-normal convolution while accounting for the prior knowledge of per cell-type gene expression profiles (see Overview at Fig. 3a). Formally, we assume $y_{ij} = \sum_t f_i^t x_{ij}^t + \epsilon_{ij}$, where $\epsilon_{ij}$ is a log-normal error with mean parameter 0 and variance parameter $\gamma_j$. Then, $x_{ij}^t$ follows a log-normal distribution: $x_{ij}^t \sim LN(\mu_j^t, \frac{1}{\lambda_j^t})$, where $\mu_j^t$ and $\lambda_j^t$ are expected value and precision in log-scale. Note that the parameters $\mu_j^t$ and $\lambda_j^t$ are shared across the samples. To incorporate prior knowledge on gene expression profiles per cell type, a hierarchical Bayesian approach was taken: $\mu_j^t$ and $\lambda_j^t$ are endowed with normal-gamma priors with hyperparameters $\mu_{0j}^t, \kappa_{0j}^t, \alpha_{0j}^t$, and $\beta_{0j}^t$: $(\mu_j^t, \lambda_j^t) \sim NG(\mu_{0j}^t, \kappa_{0j}^t, \alpha_{0j}^t, \beta_{0j}^t)$. Note that the normal-gamma distribution is a conjugate prior of log-normal distribution, based on which marginal distribution of $x_{ij}^t$ given the hyperparameters $\mu_{0j}^t, \kappa_{0j}^t, \alpha_{0j}^t$, and $\beta_{0j}^t$ is analytically tractable. The other hidden variable, $f_i^t$, was endowed with Dirichlet distribution: $(f_i^1, ..., f_i^T) \sim D(\alpha_i^1, ..., \alpha_i^T)$.

For the inference, a collapsed variational inference was employed to handle analytically intractable posterior distribution of hidden variables given observed variables[39]. In the framework, the random variables with conjugate prior distribution, which are $\mu_j^t$ and $\lambda_j^t$, were integrated out, which allows us to find a fully Bayesian estimation of $x_{ij}^t$ instead of estimation of the single most probable $\mu_j^t$ and $\lambda_j^t$[39]. By defining the variational distribution for the hidden variables, $x_{ij}^t$ and $f_i^t$, the objective function is to minimize the dissimilarity between the variational distribution and probability distribution, measured by Kullback-Leibler divergence (see Supplementary Note 2 for the detailed derivation). The minimization was done by the Limited-memory Broyden-Fletcher-Goldfarb-Shanno (L-BFGS) algorithm available in SciPy Python library with the constraints of $f_i^t \geq 0$ and $\sum_t f_i^t = 1$. Numba-compiled objective function and gradients were used for the acceleration.

**Selection of hyperparameters based on the empirical-Bayes framework.** BLADE has multiple hyperparameters for the hidden variables $x_{ij}^t$ and $f_i^t$, and also for observed variable $y_{ij}$. For $f_i^t$, a hyperparameter $\alpha_i^t$ for Dirichlet distribution needs to be set. A user-defined value is assigned to $\alpha_i^t$ for all $t$ since we do not have prior information on cellular composition. For $y_{ij}$, we need to specify a precision of each gene, $\gamma_j$, which we chose $\frac{1}{\mathbb{V}(\log y_{ij})s}$, where $s$ and $\mathbb{V}(\log y_{ij})$ are a user-defined scale factor and a variance in log-scale measured per gene, respectively. For hyperparameters of $x_{ij}^t$, $\mu_{0j}^t$, $\kappa_{0j}^t$, $\alpha_{0j}^t$, and $\beta_{0j}^t$, we incorporated prior knowledge of gene expression profiles per cell type obtained from the scRNA-seq data. Given log-normal likelihood and normal-gamma priors, average expression level and standard deviation of $x_{ij}^t$ are: $\mathbb{E}(\log x_{ij}^t) = \mu_{0j}^t$ and $\mathbb{V}(\log x_{ij}^t) = \frac{\beta_{0j}^t}{\alpha_{0j}^t}$, respectively. To make use of the prior knowledge, we obtained the sample estimates of $\mathbb{E}(\log x_{ij}^t)$ and $\mathbb{V}(\log x_{ij}^t)$ from the scRNA-seq data, denoted by $\mu_j^t$ and $(\sigma_j^t)^2$. Then, we assigned $\mu_{0j}^t = \mu_j^t$ whereas $\alpha_{0j}^t$ is set by users followed by deriving: $\beta_{0j}^t = \alpha_{0j}^t (\sigma_j^t)^2$. Here, $\alpha_{0j}^t$ allows to adapt to how much information the single cell data carries for the bulk RNA-seq data. The other hyperparameter $\kappa_{0j}^t$ is also user-defined, which serve as a scale factor for variance of $\mu_j^t$ (see also Supplementary Note 2).

An empirical Bayes approach was employed to select the best set of user-defined parameters[40]. For each configuration of parameters, a maximum likelihood estimate of variational parameters is obtained using a subset of samples. Then, the hyperparameter configuration with the highest likelihood is selected, followed by performing deconvolution using the entire data set. Only a subset of samples is used in the empirical Bayes step, not only to gain computational efficiency but also to avoid overfitting. Throughout the manuscript, we considered a total of 90 different parameter configurations that cover all possible combinations of $\alpha_i^t \in \{1, 10\}$, $\alpha_{0j}^t \in \{0.1, 0.5, 1, 5, 10\}$, $\kappa_{0j}^t \in \{1, 0.5, 0.1\}$, and $s \in \{1, 0.3, 0.5\}$.

**Construction of the simulation data with a controlled noise level.** We constructed simulation data sets of bulk gene expression profiles with known cellular fraction, gene expression profiles per cell type, and a diverse number of cell types and samples. To this end, given a number of cell types and genes, we first randomly sample an expected gene expression level $\mu_j^t$ for gene $j$ and cell type $t$ from a normal distribution with 0 mean and standard deviation of 1.5: $\mu_j^t \sim N(0, 2)$. Then, we sample gene expression levels per sample and per cell type, $x_{ij}^t$ from a log-normal distribution with mean $\mu_j^t$ and standard deviation of $\sigma$ ($x_{ij}^t \sim LN(\mu_j^t, \sigma)$), where $\sigma$ is the parameter to control the variability in gene expression per cell type of each simulation data set. Fraction of cell types are sampled from a Dirichlet distribution with uninformative prior: $f_i^t \sim (\forall_t \alpha_i^t)$, where $\alpha_i^t = 1$. Then, the bulk gene expression profiles are generated by $y_{ij} = \sum_t f_i^t x_{ij}^t$. We constructed a total of 700 training data sets with the following settings: (1) number of samples = [5,10,20,50,100]; (2) number of genes = [100,200,500,1000]; (3) number of cell types = [2,3,5,10,20]; and (4) level of variability in gene expression profiles per cell type: $\sigma = [0.1,0.2,0.5,0.75,1,1.25,1.5]$.

**Construction of PBMC simulation data.** To construct realistic simulation data, 20 bulk gene expression data sets were generated by randomly sampling and merging a subset of 9439 cells from the two PBMC scRNA-seq datasets. For each sample, the cellular fraction was first sampled from a Dirichlet distribution. The actual fractions of the 15 cell types were used as the parameter of the Dirichlet distribution so that the sampled fraction is similar to the total fraction. The fraction was then converted into the count of each cell type, with the following constraints: (1) the total number of cells is 100, and (2) the minimum number of cells per type is one. Then, the given number of cells were sampled with replacement, followed by obtaining the raw counts per cell type as the cumulative sum of the raw counts of the sampled cells. Up to three distinct cells per type were allowed to be sampled since otherwise, gene expression variability was over-stabilized due to the averaging. Finally, the simulated bulk raw counts were obtained by taking the cumulative sum of the raw counts per cell type among 15 cell types. The bulk gene expression data was log-normalized using the Seurat package[32].

**Standard bulk RNA-seq data for PBMC immune cell mixtures.** The raw counts of RNAseq data and immune cell fractions determined by flow cytometry were obtained from the GEO databases with accession GSE107572. The raw counts was log-normalized using the Seurat package[32].

**Construction of PDAC evaluation data.** PDAC single-cell RNAseq data were obtained from the Genome Sequence Archive database under the accession code CRP000653[29]. A total of 57,530 cells from 35 pancreas samples (11 normal pancreas and 24 PDAC samples) were previously classified into ten cell types. For auxiliary data, we selected 17,266 cells (30% of cells) from six samples, of which two are normal and four are PDAC samples with the most cells. The rest of the 29 samples were used as the main data for evaluation. The signature genes were selected by the top 100 DEGs from each of the ten cell types (FDR < 0.1; 818 genes in total), followed by obtaining mean and standard deviation from the reference data. Note that we used more stringent criteria to select DEGs than for the PBMC data, because a sufficient number of DEGs (>500 DEGs) still satisfies these. For main data, a cumulative sum of the raw count of all cells was obtained from each sample. The standard log-normalization was then applied to the raw count. For the evaluation, the true cell type fractions and cell-type-specific gene expression profiles were obtained per main sample.

**Systematic evaluation of BLADE and comparison against baseline methods.** The original implementation of CIBERSORTx, NNLS, and MuSiC were obtained from https://cibersortx.stanford.edu/ (docker image), SciPy Python library, and https://github.com/xuranw/MuSiC (R package), respectively. For all four methods, the same set of genes were consistently used for a fair comparison. For the simulation data sets with the controlled gene expression variability level, true mean $\mu_j^t$ and variability $\sigma$ per cell type of all genes were retrieved. For the PBMC and PDAC bulk transcriptome data, average and standard deviations of the union of DEGs of, respectively, 15 and 10 cell types were obtained from the scRNAseq data. These DEGs were selected using a FDR cutoff of 0.2 for PBMC data (in total 1007 genes) and a FDR cutoff of 0.1 for PDAC data (in total 818 genes). CIBERSORTx and NNLS require average expression profiles per gene and cell type, and BLADE

requires both mean and standard deviation. MuSiC internally calculates the gene weight using the raw counts from scRNA-seq data, which was only available in PBMC and PDAC evaluation data set. The Pearson correlation coefficient, Spearman correlation coefficient, and root mean squared error (RMSE) were measured using the predicted and true fraction of each cell type across the samples to evaluate the deconvolution performance. Likewise, the Pearson correlation coefficient was measured between true and estimated gene expression profiles per cell type for group mode purification and per cell type and per sample for the high-resolution mode purification. The performance evaluation for purification was done only for CIBERSORTx and BLADE as NNLS and MuSiC only estimate cellular fractions.

**Reporting summary**. Further information on research design is available in the Nature Research Reporting Summary linked to this article.

## Data availability
The data used in this study is from public sources. The two PBMC CITE-seq datasets of healthy donors were downloaded from 10x Genomics datasets database [https://support.10xgenomics.com/single-cell-gene-expression/datasets/3.0.2/5k_pbmc_protein_v3] [https://support.10xgenomics.com/single-cell-gene-expression/datasets/3.0.0/pbmc_10k_protein_v3]. The TCGA data is retrievable using the TCGA-biolinks R package. The PBMC data is available from GEO under the accession code GSE107572. The single-cell RNA-seq data of the PDAC cohort is available from the Genome Sequence Archive under the accession code CRP000653.

## Code availability
BLADE python software along with a user-friendly demo is available and maintained at https://github.com/tgac-vumc/BLADE[41].

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

## Acknowledgements
The authors thank Kai Ruan for his careful review of the derivation of the BLADE algorithm. This project was supported by stichting Cancer Center Amsterdam (CCA2019-9-62).

## Author contributions
Y.K. and M.W. conceived the ideas, and designed the algorithm. Y.K. developed the python software. B.A.B. and S.D.A. analyzed PDAC single-cell RNAseq and PBMC CITE-seq data. B.A.B., S.D.A., and J.G.V. classified immune cell types in the CITE-seq data. Biological interpretation of the outcome is done by S.D.A., J.O., A.F.S., J.V., F.D., H.L. B.Y., and J.G.V. Evaluation of the algorithm performance is designed and performed by Y.K. and B.A.B. All authors discussed the results and contributed to the writing.

## Competing interests
The authors declare no competing interests.
