## [Peer Review File · Nature Communications]

Bayesian log-normal deconvolution for enhanced in silico microdissection of bulk gene expression dataReviewers' Comments:

Reviewer #1:

Remarks to the Author:

In this paper, the authors present a new and promising deconvolution method for gene expression data. BLADE achieves accurate deconvolution results and also allows to estimate the gene expression profile per cell type. An important shortcoming in current deconvolution algorithms is the difficulty to handle gene expression variability without log-transformation. The log-normal convolution model of BLADE accounts for variability in gene expression, resulting in a method that can handle larger number of cell types in comparison to most other methods. In the manuscript, it is shown that BLADE outperforms CIBERSORTx (and NNLS) in deconvolution and gene expression profile estimation of each cell type.

Results, last section: A major concern on this manuscript is that only a mix of 2 scRNA-seq PBMC datasets was used to evaluate the performance of BLADE by making simulated bulk datasets. Applying the method and comparing the performance on other types of data (not PBMC) (e.g. from tissues) would improve the performance evaluation of this method. In addition, at least 1 real bulk dataset (with known cell type composition) should be included in the manuscript for method evaluation, e.g. published paired bulk and single-cell datasets.

Results, last section: It is not clear from the manuscript whether the expression data from the cells that were used as prior knowledge for deconvolution (reference dataset) were completely independent from the cells used to generate simulated bulk datasets. In other words: were cells split clearly into training and test-set. Did the authors not consider to use scRNA-seq PBMC dataset 1 as reference and PBMC dataset 2 for simulation of bulk dataset (or vice versa)? This last suggestion should be considered.

Results, last section: It would be of interest to see how the performance of BLADE compares for high abundant versus low abundant cell types in the 20 simulated datasets. What is the rationale to only generate 20 simulated bulk datasets?

Only one measure for performance analysis, i.e. Pearson correlation was used. Do other measures (e.g. mean squared error) give similar results/conclusions?

Results, section 2: It is not clear why EPIC is used as deconvolution method to compare the probabilistic assumptions.

Figure legend 2: b and c are switched.

Reviewer #2:

Remarks to the Author:

Review of "BLADE: Bayesian Log-normAI Deconvolution for enhanced in silico microdissection of bulk gene expression data" by arbosa et al

This paper develops a Bayesian deconvolution procedure for bulk gene expression data utilizing single-cell RNA-seq as prior knowledge. In contrast to existing deconvolution approaches, the proposed approach models the inherently variable nature of gene expression under the probabilistic framework and estimates both cellular make-up and gene expression profiles of each cell type in each sample. Computationally, an efficient variational inference has been proposed so data with a large number of cell types can be analyzed. Given these features, I believe BLADE will be a useful addition to the deconvolution tools, in particular for unraveling heterogeneous cellular activity in complex biological systems. However, given these good properties, I do have some serious concerns on the

paper, which I include below:

1. I'm really troubled by using maximal log-likelihood, or equivalently AIC/BIC (given the same # of parameters) as a way to compare across models from different distributions, since the underlying formulas are different and may not be comparable. The log-normal model consistently gives the highest maximal log-likelihood (Fig 2a) but this may not guarantee that it's the best candidate distribution for the data.
2. Practically, I'm not sure if deconvoluting bulk expression data up to >20 cell types is a good idea. Given a large number of cell types, some cell types are likely to be very similar in their gene expression which can lead to the so called collinearity issue, and thus unreliable cell composition estimates. Fig 5a shows that as the # of cell types increases, BLADE can have a very wide performance range. The authors need to further investigate and propose some practical guidelines on when and how to merge similar cell types, for examples. By the way, Figure 5 is very crowded and some sub-figures need to be deleted, such as Fig5 d-e. The same is true for Figure 6.
3. MuSiC does weight genes based on their expression variabilities, and should be included as a comparison approach.
4. Can the authors confirm if sigma (line 1 page 19) is set the same for all genes or gene dependent? Real data often suggests the second situation.
5. Fig 5b, why there is a systematic dip at the variability of 0.2?
6. Setting the # of cell to 100 (line 17 page 19) might be too low.
7. It is mentioned that "Among the 15 cell types, plasmablasts and classical/nonclassical monocytes were the best predicted by all three methods, whereas the methods commonly failed to predict the composition of regulatory T-cells (Tregs), naive CD8+ T-cells (NaiveCD8T), and plasmacytoid dendritic cells (pDC). These poorly predicted cell types were low abundant (less than 2%; Supplementary Fig. S7), indicating the difficulty in deconvolution of rare cell populations. However, some of the low abundant cell types were well-predicted, such as plasmablasts, and thus the abundance is not the sole determinant of performance." It would be interesting to investigate in addition to the cell abundance, what factors affect the deconvolution performance.

Some example grammar errors etc:

1. Line 4 from the bottom of page 9, "data The" please add . between the two words.
2. Lines 3-4 of the 2nd paragraph on page 17, "for which we chose one value across the different t s since we do not have prior information on cellular composition", please remove the space between t and s or change "t s" to "t values"
3. Page 25, line 6, change "differentially expressed genes (red) and non-differentially expressed genes (DEG; blue)" to "differentially expressed genes (DEG; red) and non-differentially expressed genes (blue)".

Reviewer #3:

Remarks to the Author:

In this study, Barbosa et al. introduce a new computational method to deconvolve bulk gene expression using single cell RNA-Seq datasets – BLADE. BLADE is able to estimate cell type proportions and gene expression. The novelty of BLADE, over other methods, is that it uses a Bayesian framework and assuming log-normal distribution instead of normal to better capture the variance in gene expression. In addition, BLADE is able to handle over 20 cell types. Overall, BLADE outperforms existing methods both in estimating cell type proportions and expression profiles (outputting more genes and more accurately). I have concerns regarding the biological novelty that can be achieved using BLADE which the authors can demonstrate better and some minor concerns regarding the flow of the manuscript.

Major comments:

- Previous studies have shown that gene expression follows Poisson distribution (for example: Grun, Kester, and Oudenaarden 2014; Klein et al. 2015), especially scRNA-Seq. I suggest that the authors include this distribution in their analysis as well and compare it to the other distributions examined in the manuscript.
- Figure 2c – this analysis can be generalized for all genes? Seeing only two genes is nice because you can understand the point the authors are trying to make, but I am afraid it might not represent all genes. For example, is there any dependency to the expression level of the gene?
- The simulated data is done very nicely but I think it would be also interesting to use real bulk RNA-Seq PBMC data that we know the true fractions of each cell type using an independent method such as FACS measurements (for example the validation cohort of bulk RNA-Seq generated in the ciphersortX paper - GEO: GSE127813 or any other that is available). Also, why only 100 cells were used to create the mixtures? Seems like it's a small number of cells that does not represent real bulk data that usually has many more cells.
- What happens if you one of the cell-types in the bulk dataset is missing from the single cell data? The authors discuss this a bit in the discussion ("Furthermore, BLADE may be beneficial in handling cell types without a precise prior knowledge". "For instance, BLADE can be applied to estimate gene expression profiles of each cell type that makes up the tumor microenvironment (TME).") but I think this point can be tested further. Sometimes, the dissociation to single cells done prior to scRNA-Seq can lead to depletion of some of the cells, for example, adipocytes, and therefore can lead to differences between bulk and single-cell RNA-Seq.
- One of the things that is missing from the manuscript is a demonstration of what kind of biological novelty can be achieved with this method. In the discussion you mention TME in PDAC, can this be investigated more? Another option, can this method be used to now identify new subtypes of tumor types?

Minor comments:

- Figures management - there are several issues with the figures that interrupt the flow of the manuscript:
 - o Captions font should be bigger and better defined. Some examples: 'Ngene' is not a phrase that will be clear for all readers, titles of panel in figure 1a have typos, legends in figure 1 are very small, figure 3 axis labels are not clear and should describe better what is shown, figure S6 (right instead of right) and the caption is not finished, the titles and captions of figure 2 and 4 include additional text that should be in the main text, figure 1d x-axis tick labels are mixed up (there is twice 0.5). There are more typos and in general, the figures need work.
 - o Merging figure: Figure 3 can be merged with another figure or in the supplementary material. Figure 4 and 5 can be merged.
 - o Figure S4 quality is bad and should be enhanced.
 - o Figure 5 has a lot panels and is very busy which makes it really hard to read. For example, panel b: maybe it will be better to show one condition for group mode purification and one for high-resolution purification. That way, the graphs can be bigger, and the reader will be able to see the differences. The rest of the panels can be shown in the supplementary. In addition, panels d and e – the colorbar direction is confusing and maybe can be inverse. Also, can we see the same analysis for BLADE?
- Methods - PBMC single-cell RNA-seq data – what is the criteria for DE genes? What was the FDR? fold change? Or any other metric that was used.
- Pearson correlation should state coefficient.

We like to thank all the reviewers for their constructive comments. We further build on our manuscript based on these comments. In particular, on the following aspects:

- We evaluated the log-normal distribution based on RMSE, in addition to the log-likelihood as in the previous manuscript. As described below, we think the maximum-likelihood approach is fair since the same number of parameters are used in a log-normal, normal, and negative binomial distribution. And we observed a similar outcome with the RMSE, where log-normal performs slightly better than the negative binomial. The negative binomial also performs comparably well, but we chose the log-normal over negative binomial since the log-normal assumption is more tractable in the Bayesian framework.
- We found a small error in the derivation of the BLADE algorithm. We updated the algorithm accordingly and performed experiments using the new algorithm. The correction in the algorithm resolved one of the issues reviewers mentioned, the slight dip in the performance of algorithms. The latest version of python package is also available.
- One extra method (MuSiC), a variant of NNLS that accounts for inter-subject variability, was introduced for the comparison as suggested. Note that since MuSiC explicitly requires single-cell data, we could not run MuSiC on simulation data that lacks single-cell data.
- For the PBMC experiment, we introduced a simpler classification of immune cells to diversify the difficulty of deconvolution (4 different levels). It allows us to study the influence of the number of cell types in the deconvolution performance. We confirmed that all algorithms performed well with a smaller number of cell types (e.g., with four cell types, all algorithms reached > 0.5 Pearson correlation for estimating cell type fractions; Figure 5). We also noted that BLADE could achieve high performance with 15 cell types; however, not for all cell types. We found that the cell types with low performance are less abundant and have fewer differentially expressed genes. These experiments will help readers to choose the number of cell types for deconvolution.
- We performed extra experiments using 36 scRNA-seq data from PDAC samples. For this experiment, no sampling was done to simulate bulk gene expression data but instead we pulled all cells per sample. Furthermore, we split auxiliary (7 samples) and main data sets (29 samples) in which only auxiliary data sets were used to derive signatures. This experiment is the most realistic data for evaluation, in which we confirmed the novelty and performance of BLADE.

We will reply to the reviewer comments one by one.

Reviewer #1 (Expertise: Deconvolution of Bulk RNASeq data):

In this paper, the authors present a new and promising deconvolution method for gene expression data. BLADE achieves accurate deconvolution results and also allows to estimate the gene expression profile per cell type. An important shortcoming in current deconvolution algorithms is the difficulty to handle gene expression variability without log-transformation. The log-normal convolution model of BLADE accounts for variability in gene expression, resulting in a method that can handle larger number of cell types in comparison to most other methods. In the manuscript, it is shown that BLADE outperforms CIBERSORTx (and NNLS) in deconvolution and gene expression profile estimation of each cell type.

Results, last section: A major concern on this manuscript is that only a mix of 2 scRNA-seq PBMC datasets was used to evaluate the performance of BLADE by making simulated bulk datasets. Applying the method and comparing the performance on other types of data (not PBMC) (e.g. from tissues) would improve the performance evaluation of this method. In addition, at least 1 real bulk dataset (with known cell type composition) should be included in the manuscript for method evaluation, e.g. published paired bulk and single-cell datasets.

- **Thanks for the suggestion. In the revised manuscript, we expanded our analysis further by including single-cell RNA-seq data from 36 pancreatic tissue samples. For this experiment, we split the training (7 samples) and test data set (29 samples) so that we can make the performance evaluation in a fair manner. Furthermore, no sampling was involved but instead obtained a cumulative count per gene and per sample (see Figure 6).**

Relevant part in the revised manuscript:

- subsection *Evaluation of BLADE for deconvolution of tumor RNA-seq data* of Result section.

Results, last section: It is not clear from the manuscript whether the expression data from the cells that were used as prior knowledge for deconvolution (reference dataset) were completely independent from the cells used to generate simulated bulk datasets. In other words: were cells split clearly into training and test-set. Did the authors not consider to use scRNA-seq PBMC dataset 1 as reference and PBMC dataset 2 for simulation of bulk dataset (or vice versa)? This last suggestion should be considered.

- **The training and test were not split in PBMC cases. This is not done since we intended to evaluate the algorithms when prior knowledge of cell-type specific gene expression profiles is perfect. Note that though the evaluation is still fair since the same prior knowledge (or signatures) were given to all the algorithms evaluated in the manuscript. In the revised manuscript, we kept the PBMC data as is, and revised in the text to make it clear that all cells were used to create the prior knowledge. In addition, we introduced PDAC datasets for the further evaluation, where samples were split into training and test data. We like to**

emphasize that the same prior knowledge was used for all the algorithms used in the study.

Relevant part in the revised manuscript:

- In *Construction of PBMC simulation data* of Methods section:

“To construct realistic simulation data, 20 bulk gene expression data sets were generated by randomly sampling and merging a subset of 9,439 cells from the two PBMC scRNA-seq datasets.”

- In the subsection *Evaluation of BLADE for deconvolution of tumor RNA-seq data* in Results section.:

“For a fair evaluation of deconvolution algorithms, the 36 samples and their cells were split into reference (nine samples, of which five are tumors) and main samples (27 samples, of which 20 are tumors; **Supplementary Fig. S22**).”

- In the subsection *Application of BLADE to in silico mixture of PBMC scRNA-seq data* in Results section:

“Using the bulk PBMC data generated above, we evaluated BLADE taking CIBERSORTx, NNLS, and also MuSiC as the baseline. We used the same list of genes and signatures for the baseline methods for a fair comparison.”

Results, last section: It would be of interest to see how the performance of BLADE compares for high abundant versus low abundant cell types in the 20 simulated datasets. What is the rationale to only generate 20 simulated bulk datasets?

- We compared the performance between the cell type abundance and the number of unique DEGs with deconvolution performance (Figure 5d, 6d, and Supplementary Figs. S17). There is a trend where a subset of low-abundant cells with a small number of unique DEGs achieved low performance. Furthermore, from the simulation experiment (Figure 4, Supplementary Fig. S6), we could observe, in general, lower performance for the simulation data set with many cell types. This trend is consistent between the four methods used in the study. We think this can help readers to decide the number of cell types for deconvolution.
- We chose to generate 20 bulk datasets for PBMC as this could give us rather accurate estimation of the performance (e.g., 95% Confidence interval of 0.7 Pearson correlation is 0.37 and 0.87). Furthermore, we have multiple cell types to robustly estimate the performance. In the simulation data with varying sample size (n=5 to 100), we did not observe significant difference in the estimated performance between the datasets (Supplementary Fig. S8-9).

Relevant part in the revised manuscript:

- In *Application of BLADE to in silico mixture of PBMC scRNA-seq data* of Results section: “Among the 15 cell types, plasmablasts, classical monocytes, NK cells were the best

predicted by all four methods, which commonly failed to predict the composition of regulatory T cells (Tregs), naive CD8⁺ T cells (NaiveCD8T), and central memory CD4⁺ T cells (CMCD4T). These cell types are commonly low abundant (fraction of <7% on average), and only a few unique DEGs were identified for each cell type (< 50 unique DEGs; **Fig. 5d**; see **Supplementary Fig. S17** for other levels)."

- In *Evaluation of BLADE for deconvolution of tumor RNA-seq data* of Results section: "Most cell types achieved high performance (>0.5 of Pearson correlation coefficient) in all methods, except for B cells (in MuSiC and CIBERSORTx), T cells (in CIBERSORTx and NNLS), and Stellate cells (in NNLS). These cell types are often less dominant (less than <5%) and with a small number of DEGs (less than 40 unique DEGs; **Fig. 6d**)."

Only one measure for performance analysis, i.e. Pearson correlation was used. Do other measures (e.g. mean squared error) give similar results/conclusions?

- **As by your suggestion, we now measure RMSE to complement the Pearson correlation coefficient. However, RMSE is very focused on the estimation of the abundant cell types, as the small fractions contribute very little. In fact, we noticed that when increasing the number of cell types (which practically leads to a decrease of abundances), the RMSE always improved. See Supplementary Figures S8 and S16 for the trend. This is quite misleading as the increasing number of cell types should make the problem more complicated. We included RMSE outcome (Supplementary Figs. S8, S16, and S23), but the discussion is still focused on the Pearson correlation coefficient, and we also placed a note in the legends that RMSE is not meant to be compared between datasets. We acknowledge that an additional performance metric is useful, so we also added Spearman's rank correlation coefficients (Supplementary Figs. S7, S16, S23), which is mostly consistent with the Pearson correlation coefficients.**

Relevant part in the revised manuscript:

- Supplementary Figures S7-8, S16, and S23.

- In the legend of Supplementary Fig. S7 and S8:

"Note that RMSE is not meant to be compared between data set with the different number of cell types, as it depends a lot on the abundance of cell types. (according to RMSE, the performance gets better with the higher number of cell types, which is misleading)."

Results, section 2: It is not clear why EPIC is used as deconvolution method to compare the probabilistic assumptions.

- **The point of this part of the manuscript is to evaluate which statistical models can account for variability of gene expression pattern per cell type. For this, we need to provide cell type fractions, unlike Figure 2 a-c where the evaluation was done for bulk transcriptome. We chose EPIC for this since it was**

previously reported for deconvolution of TCGA data and is not part of the baseline methods. We revised the text to make this point clear.

Relevant part in the revised manuscript:

In *Modeling gene-expression variability by probabilistic distribution* of Results section: "First, we obtained TCGA RNA-seq data of mesothelioma (TCGA-MESO; n=84) and sarcoma (TCGA-SARC; n=256), from which we estimated the fraction of eight cell types using EPIC¹⁷, a deconvolution method previously applied to the TCGA."

Figure legend 2: b and c are switched.

- Thank you so much for detecting the error. We corrected the legends as such.

Reviewer #2 (Expertise: Deconvolution of Bulk RNASeq data):

Review of "BLADE: Bayesian Log-normal AI Deconvolution for enhanced in silico microdissection of bulk gene expression data" by arbosa et al

This paper develops a Bayesian deconvolution procedure for bulk gene expression data utilizing

single-cell RNA-seq as prior knowledge. In contrast to existing deconvolution approaches, the proposed approach models the inherently variable nature of gene expression under the probabilistic framework and estimates both cellular make-up and gene expression profiles of each cell type in each sample. Computationally, an efficient variational inference has been proposed so data with a large number of cell types can be analyzed. Given these features, I believe BLADE will be a useful addition to the deconvolution tools, in particular for unraveling heterogeneous cellular activity in complex biological systems. However, given these good properties, I do have some serious concerns on the paper, which I include below:

1. I'm really troubled by using maximal log-likelihood, or equivalently AIC/BIC (given the same # of parameters) as a way to compare across models from different distributions, since the underlying formulas are different and may not be comparable. The log-normal model consistently gives the highest maximal log-likelihood (Fig 2a) but this may not guarantee that it's the best candidate distribution for the data.

- In principle, it is widely believed that, unlike the likelihood ratio-test, AIC can be used to compare non-nested models (as in our case). In fact, Akaike himself phrased this in his seminal AIC-paper [1]: "One important observation about AIC is that it is defined without specific reference to the true model [$f(x|k\theta)$]. Thus, for any finite number of parametric models, we may always consider an extended model that will play the role of [$f(x|k\theta)$] This suggests that AIC can be useful, at least in principle, for the comparison of models which are nonnested, i.e., the situation where the conventional log likelihood-ratio test is not applicable."

In our case, comparing AICs of the two models is equivalent to comparing the log-likelihoods, as the models have the same number of parameters. Nevertheless, we do agree it may be good to consider an alternative metric as well. By your suggestions, we have therefore also compared the two models in terms of root Mean Squared Error (with data and predictions on log-scale). We found that these also largely agree for the two models (see Supplementary Fig. S4).

[1] Akaike, H. "Prediction and entropy." Selected Papers of Hirotugu Akaike. Springer, New York, NY, 1985. 387-410.

Relevant part in the revised manuscript:

In *Modeling gene-expression variability by probabilistic distribution* of Results section: "In terms of log-likelihood measured per gene, log-normal and negative binomial deconvolutions performed equally well for most of the genes, except for a few genes with a more favorable performance with log-normal (**Fig. 2d** and **Supplementary Fig. S4**)."

2. Practically, I'm not sure if deconvoluting bulk expression data up to >20 cell types is a good idea. Given a large number of cell types, some cell types are likely to be very similar in their gene expression which can lead to the so called collinearity issue, and thus unreliable cell composition estimates. Fig 5a shows that as the # of cell types increases, BLADE can have a very wide performance range. The authors need to further investigate and propose some practical guidelines on when and how to merge similar cell types, for examples.

- We appreciate this comment from the reviewer. We performed extra experiments with the PBMC dataset with a coarse classification of immune cells to reduce the number of cell types. In total, we introduced four different levels of immune cell classification. As the reviewer pointed out, we could see that the performance gets worse as the number of cell types gets higher, particularly for the lowly abundant cell types and those with small numbers of differentially expressed genes. This information can provide an insight to the reader on how to select the number of cell types.

Relevant part in the revised manuscript:

- In *Application of BLADE to in silico mixture of PBMC scRNA-seq data* of Results section: "We also generated three extra data sets with a coarse classification of the 15 cell types by four (level 1; 441 genes selected), eight (level 2; 604 genes), and 12 cell types (level 3; 880 genes) in the same manner to diversify the difficulty levels for deconvolution (see **Supplementary Table. 1** for the details of classifications)."

- In *Application of BLADE to in silico mixture of PBMC scRNA-seq data* of Results section: "Among the 15 cell types, plasmablasts, classical monocytes, NK cells were the best predicted by all four methods, which commonly failed to predict the composition of regulatory T cells (Tregs), naive CD8⁺ T cells (NaiveCD8T), and central memory CD4⁺ T cells

(CMCD4T). These cell types are commonly low abundant (fraction of <7% on average), and only a few unique DEGs were identified for each cell type (< 50 unique DEGs; **Fig. 5d**; see **Supplementary Fig. S18** for other levels).”

- In *Evaluation of BLADE for deconvolution of tumor RNA-seq data* of Results section: “Most cell types achieved high performance (>0.5 of Pearson correlation coefficient) in all methods, except for B cells (in MuSiC and CIBERSORTx), T cells (in CIBERSORTx and NNLS), and Stellate cells (in NNLS). These cell types are often less dominant (less than <5%) and with a small number of DEGs (less than 40 unique DEGs; **Fig. 6d**).”

By the way, Figure 5 is very crowded and some sub-figures need to be deleted, such as Fig5 d-e. The same is true for Figure 6.

- **We simplified the Figures 5 (Figure 4 in the revised manuscript) in the main by showing the result from a subset of the simulation data set while retaining them in the supplementary information. Also, we replace the radar plot in Figure 6 (Figure 5 in the revised manuscript) with a simpler visualization (boxplot). A consistent visualization was done across the Figures so that readers can follow easily.**

3. MuSiC does weight genes based on their expression variabilities, and should be included as a comparison approach.

We really appreciate this comment. MuSiC does take into account gene expression variabilities, which practically weigh genes on their importance for deconvolution. As suggested, we included MuSiC in our evaluation using PBMCs and PDAC data. Note we do not have MuSiC for simulation data experiments as they explicitly require raw counts from the single-cell data. In our evaluation (Figures 5-6), MuSiC performed well, slightly better than BLADE in several data sets, including the PDAC data set. In particular, we noted a high correlation between MuSiC and BLADE, likely due to the gene expression variability commonly accounted for. Taken together with CIBERSORTx, it is clear that linear regression-based deconvolution can perform significantly better by focusing on a subset of genes, at least for fraction estimation. CIBERSORTx selects the genes that are difficult to reconstruct (support vector regression) while MuSiC weighs genes based on their known variability. However, this strategy reduces the completeness in purification, and in fact, a significant proportion of genes were left out by CIBERSORTx, which performs the purification using a linear regression method. Note that MuSiC does not perform purification at all. Furthermore, the Bayesian framework of BLADE naturally estimates the uncertainty in the prediction outcome, which is not possible with MuSiC. We think the novelty of BLADE is more apparent thanks to this experiment. We revised the discussion session to make this point clear.

Relevant part in the revised manuscript:

- In *Application of BLADE to in silico mixture of PBMC scRNA-seq data* of Results section:

“Using the bulk PBMC data generated above, we evaluated BLADE taking CIBERSORTx, NNLS, and also MuSiC as the baseline”

“MuSiC is an exception where the performance gets higher from level 1 to 3. At level 4, BLADE outperformed CIBERSORTx (P-value of 0.0087; a one-tailed paired t-test) and NNLS (P-value of 0.021; a one-tailed paired t-test) and performed comparably to MuSiC (P-value of 0.46; one-tailed paired t-test).”

- In *Evaluation of BLADE for deconvolution of tumor RNA-seq data* of Results section:

“For predicting the fraction of 10 cell types, MuSiC performed the best, followed by BLADE and CIBERSORTx (**Fig. 6b**; see Spearman correlation coefficients in **Supplementary Fig. S23**). Interestingly, the performance of BLADE correlates the most with MuSiC (Pearson correlation coefficient of 0.62; P-value of 0.056), whereas it is less so with CIBERSORTx (Pearson correlation coefficient of 0.39; P-value of 0.27) and NNLS (Pearson correlation coefficient of -0.18; P-value of 0.62; **Fig. 6c**).”

- In Discussion section:

“CIBERSORTx and MuSiC are also linear-regression approaches that partially alleviate the issue by prioritizing genes for deconvolution. Support vector regression, the core algorithm of CIBERSORTx, depends on a subset of genes with high reconstruction errors. On the contrary, MuSiC explicitly learns gene weights from the single-cell RNA-seq data and prioritizes genes with low variability, which likely reduces skewness and hence improves accuracy of the normal distribution.”

“MuSiC outperformed BLADE in several cases, indicating normal distribution-based deconvolution can also be accurate if genes are prioritized based on the gene expression variability. However, the strategy of prioritizing genes reduces the completeness of the purification results (**Figs. 4d, 5f, 6g**).”

“This similarity led to a highly correlated performance between BLADE and MuSiC (**Figs. 5c and 6c**), while BLADE also performs the purification. Furthermore, the Bayesian framework of BLADE allows estimation of the uncertainty in the prediction, which may be valuable to evaluate the quality of the results.”

4. Can the authors confirm if sigma (line 1 page 19) is set the same for all genes or gene dependent? Real data often suggests the second situation.

- **Thanks for the interesting point. For PBMC and PDAC data indeed have a different variability per gene. In simulation data, sigma is one of the parameters we set per data set (i.e., fixed for all genes). We chose to set this because we wanted to evaluate the impact of gene expression variability in deconvolution performance. By having smaller sigma values for a subset of genes, a deconvolution algorithm may still perform well based on the small subset. We think this way, it is clearer to understand how gene expression variability can impact the deconvolution performance. Besides, we have several extra evaluation data sets (PBMC and PDAC dataset) that are more realistic than the simulation data.**

Relevant part in the revised manuscript:

- In *Construction of the simulation data with a controlled noise level* of Methods section:

"Then, we sample gene expression levels per sample and per cell type, x_{ij}^t from a log-normal distribution with mean μ_j^t and standard deviation of σ ($x_{ij}^t \sim LN(\mu_j^t, \sigma)$), where σ is the parameter to control the variability in gene expression per cell type of each simulation data set."

5. Fig 5b, why there is a systematic dip at the variability of 0.2?

- **We also noticed that BLADE sometimes performs less well on the data set with low variability. This was due to the error in the derivation, which impacts more when the variability is low. In the revised manuscript where corrected BLADE was used, the systematic dip does not appear anymore (see Supplementary Figures S8-9).**

6. Setting the # of cell to 100 (line 17 page 19) might be too low.

- **We agree with the reviewer that 100 cells per bulk data may seem low. We also initially sampled more cells to simulate the bulk gene expression data. However, when we sample more cells per sample, there are more cells commonly sampled in 20 simulated bulk samples, which resulted in low variability in cell-type-specific gene expression profiles between these samples. We decided to keep it low to maintain the variability also in the revised manuscript (see Supplementary Fig. S12). Instead, the newly introduced pancreatic data set in the revised manuscript used all cells per sample to obtain bulk profiles, so this should serve as the most realistic evaluation data set.**

Relevant part in the revised manuscript:

- In *Application of BLADE to in silico mixture of PBMC scRNA-seq data* of Results section:

"We chose to use 100 cells as we sample more, more cells get commonly selected in multiple samples, making the simulated bulk gene expression data lose variability between the samples."

"The resulting simulation data recapitulate the gene expression variability of 15 cell types (**Fig. 5a; Supplementary Fig. S13**)."

7. It is mentioned that "Among the 15 cell types, plasmablasts and classical/nonclassical monocytes were the best predicted by all three methods, whereas the methods commonly failed to predict the composition of regulatory T-cells (Tregs), naive CD8+ T-cells (NaiveCD8T), and plasmacytoid dendritic cells (pDC). These poorly predicted cell types were low abundant (less than 2%; Supplementary Fig. S7), indicating the difficulty in deconvolution of rare cell populations. However, some of the low abundant cell types were well-predicted, such as plasmablasts, and thus the abundance is not the sole determinant of performance." It would be interesting to investigate in addition to the cell

abundance, what factors affect the deconvolution performance.

- **As stated earlier, we performed an extra analysis with PBMC and pancreatic data sets and confirmed that cell abundance and the number of unique differentially expressed genes could influence the deconvolution performance. The results are now presented in Figure 5d, Figure 6d, and Supplementary Fig. S17.**

Relevant part in the revised manuscript:

- In *Application of BLADE to in silico mixture of PBMC scRNA-seq data* of Results section: “Among the 15 cell types, plasmablasts, classical monocytes, NK cells were the best predicted by all four methods, which commonly failed to predict the composition of regulatory T cells (Tregs), naive CD8⁺ T cells (NaiveCD8T), and central memory CD4⁺ T cells (CMCD4T). These cell types are commonly low abundant (fraction of <7% on average), and only a few unique DEGs were identified for each cell type (< 50 unique DEGs; **Fig. 5d**; see **Supplementary Fig. S18** for other levels).”
- In *Evaluation of BLADE for deconvolution of tumor RNA-seq data* of Results section: “Most cell types achieved high performance (>0.5 of Pearson correlation coefficient) in all methods, except for B cells (in MuSiC and CIBERSORTx), T cells (in CIBERSORTx and NNLS), and Stellate cells (in NNLS). These cell types are often less dominant (less than <5%) and with a small number of DEGs (less than 40 unique DEGs; **Fig. 6d**).”

Some example grammar errors etc:

1. Line 4 from the bottom of page 9, “data The” please add . between the two words.
2. Lines 3-4 of the 2nd paragraph on page 17, “for which we chose one value across the different t s since we do not have prior information on cellular composition”, please remove the space between t and s or change “t s” to “t values”
3. Page 25, line 6, change “differentially expressed genes (red) and non-differentially expressed genes (DEG; blue)” to “differentially expressed genes (DEG; red) and non-differentially expressed genes (blue)”.

- **All the grammars errors were corrected, thanks for detecting them.**

Reviewer #3 (Expertise: Deconvolution of Bulk RNASeq data):

In this study, Barbosa et al. introduce a new computational method to deconvolve bulk gene expression using single cell RNA-Seq datasets – BLADE. BLADE is able to estimate cell type proportions and gene expression. The novelty of BLADE, over other methods, is that it uses a Bayesian framework and assuming log-normal distribution instead of normal to better capture the variance in gene expression. In addition, BLADE is able to handle over 20 cell types. Overall, BLADE outperforms existing methods both in estimating cell type proportions and expression profiles (outputting more genes and more accurately). I have concerns regarding the biological novelty that can be achieved

using BLADE which the authors can demonstrate better and some minor concerns regarding the flow of the manuscript.

Major comments:

- Previous studies have shown that gene expression follows Poisson distribution (for example: Grun, Kester, and Oudenaarden 2014; Klein et al. 2015), especially scRNA-Seq. I suggest that the authors include this distribution in their analysis as well and compare it to the other distributions examined in the manuscript.

- **Indeed, researchers have shown that the Poisson distribution can adequately model technical variability for both RNAseq and scRNAseq. However, it does generally not suffice for modelling biological variability (e.g. between individuals), which is why the negative binomial (NB) is often used for analysing sequencing data as an important extension of the Poisson, allowing for more dispersion [see e.g. Anders et al. (2013)]. As the Poisson distribution is a member of the NB family it can not outperform the NB in terms of fit to the data (because the NB has one more extra parameter), so therefore we feel the comparison between NB and log-normal suffices. For the sake of clarity we mention now that the negative binomial includes the Poisson distribution.**

Anders, S., McCarthy, D. J., Chen, Y., Okoniewski, M., Smyth, G. K., Huber, W., & Robinson, M. D. (2013). Count-based differential expression analysis of RNA sequencing data using R and Bioconductor. *Nature protocols*, 8(9), 1765.

Relevant part in the revised manuscript:

In *Modeling gene-expression variability by probabilistic distribution* of Results section:

“Note that Poisson distribution was also introduced for modeling count data^{23,24}, but it is a special case of negative binomial.”

- Figure 2c – this analysis can be generalized for all genes? Seeing only two genes is nice because you can understand the point the authors are trying to make, but I am afraid it might not represent all genes. For example, is there any dependency to the expression level of the gene?

- **We performed this analysis with all genes, but then in a quantitative manner by comparing maximum likelihood. The scatter plots in Figure 2c in the revised manuscript compares the maximum-likelihood between lognormal, normal, and normal distribution for all genes. In general, normal distribution tends to be lower in likelihood. The examples in Figure 2b are those with a lower likelihood for normal distribution than other distributions. We clarify this point in the revised manuscript.**

Relevant part in the revised manuscript:

In *Modeling gene-expression variability by probabilistic distribution* of Results section:

“The log-normal distribution, in general, shows the best performance in per-gene maximum likelihood, followed by the negative binomial and normal distributions (**Figs. 2a-c**). In particular, we noted a biased fit of the normal distribution towards outlier observations, in contrast to the log-normal and negative binomial distribution (see four genes with a lower maximum likelihood for normal distribution in **Fig. 2b**).”

- The simulated data is done very nicely but I think it would be also interesting to use real bulk RNA-Seq PBMC data that we know the true fractions of each cell type using an independent method such as FACS measurements (for example the validation cohort of bulk RNA-Seq generated in the cibersortX paper - GEO: GSE127813 or any other that is available). Also, why only 100 cells were used to create the mixtures? Seems like it's a small number of cells that does not represent real bulk data that usually has many more cells.

- **Taking multiple suggestions from the reviewers, we expanded our analysis by introducing multi-sample single-cell RNA-seq data from multiple subjects. We chose this data as the major novelty of BLADE is the purification, and for evaluation of the performance, we need single-cell RNAseq data. We also explored the datasets from the CIBERSORTx paper as the reviewer suggested, but we could not find a good data set that can be used to evaluate purification and fraction estimation. We did not sample cells for the pancreatic data set to simulate bulk data but then pull all cells per sample to retain real inter-sample variability. We think this is the best evaluation data set we could find to evaluate the purification performance.**

- What happens if you one of the cell-types in the bulk dataset is missing from the single cell data? The authors discuss this a bit in the discussion (“Furthermore, BLADE may be beneficial in handling cell types without a precise prior knowledge”. “For instance, BLADE can be applied to estimate gene expression profiles of each cell type that makes up the tumor microenvironment (TME).”) but I think this point can be tested further. Sometimes, the dissociation to single cells done prior to scRNA-Seq can lead to depletion of some of the cells, for example, adipocytes, and therefore can lead to differences between bulk and single-cell RNA-Seq.

- **We completely agree with the reviewer that it would be important to acknowledge that some cell types may not be covered by single-cell RNA-seq data. Also, BLADE is capable of reconstructing cell-type-specific gene expression profiles and including cell types without good prior knowledge. However, to establish this, we need to design a separate study on how to provide non-informative prior information for missing cell types. Furthermore, none of the baseline methods allows for the handling of cell types without any prior knowledge. Thus, we consider it as a future project. Note that the phrases pointed out by the reviewer are now replaced with a more detailed comparison between BLADE and MuSiC/CIBERSORTx.**

- One of the things that is missing from the manuscript is a demonstration of what kind of biological novelty can be achieved with this method. In the discussion you mention TME in PDAC, can this be investigated more? Another option, can this method be used to now identify new subtypes of tumor types?

- **This is a great point and an obvious next step with BLADE. The novel biology we can learn by the application of BLADE is from the estimated cell-type-specific gene expression profiles. Given the cell-type-specific gene expression profiles, we can identify each cell type's subtype and characterize distinct pathway activities in each cell type. We could also characterize molecular subtypes defined by bulk gene expression profiles, for instance, transcriptome-based PDAC subtypes previously reported. We clarified these points in the last paragraph of the discussion section.**

Relevant part in the revised manuscript:

In Discussion section:

“Enhanced *in silico* microdissection by BLADE opens up the possibility to molecularly characterize individual cell types in tissue based on the standard RNA-seq data. For instance, we demonstrated that BLADE could be applied to estimate each cell type's gene expression profiles that make up the tumor microenvironment (TME). This allows us to characterize pathway activity in each immune cell type and possibly to recognize additional cell (sub-)types. Furthermore, BLADE can aid previously established gene expression subtypes (e.g., PDAC^{28,29}) by characterizing the subtypes with distinct TME profiles. Finally, the detailed profiling of the TME, particularly immune TME profiles, may lead to a clinically applicable biomarker strategy for immunotherapy based on the standard bulk gene expression profiling.”

Minor comments:

- Figures management - there are several issues with the figures that interrupt the flow of the manuscript:

- o Captions font should be bigger and better defined. Some examples: 'Ngene' is not a phrase that will be clear for all readers, titles of panel in figure 1a have typos, legends in figure 1 are very small, figure 3 axis labels are not clear and should describe better what is shown, figure S6 (right instead of fright) and the caption is not finished, the titles and captions of figure 2 and 4 include additional text that should be in the main text, figure 1d x-axis tick labels are mixed up (there is twice 0.5). There are more typos and in general, the figures need work.

- All of above comments are addressed in the revised figures.

- o Merging figure: Figure 3 can be merged with another figure or in the supplementary material. Figure 4 and 5 can be merged.

- Figure 3 is now merged with Figure 2. We kept Figures 4 and 5 (3 and 4 in the revised manuscript) separate, as they looked too crowded when merged.

- o Figure S4 quality is bad and should be enhanced.

- **The quality is enhance in the revised Figure.**

o Figure 5 has a lot panels and is very busy which makes it really hard to read. For example, panel b: maybe it will be better to show one condition for group mode purification and one for high-resolution purification. That way, the graphs can be bigger, and the reader will be able to see the differences. The rest of the panels can be shown in the supplementary. In addition, panels d and e – the colorbar direction is confusing and maybe can be inverse. Also, can we see the same analysis for BLADE?

- **We made a selection for Figure 5 and now present only a subset of panels as the reviewer suggested. All the information is still available in the Supplementary Information. The color bar direction has changed as suggested. Note that BLADE does not filter any gene for purification, so the missing value analysis is not necessary.**

• **Methods - PBMC single-cell RNA-seq data – what is the criteria for DE genes? What was the FDR? fold change? Or any other metric that was used.**

- **We selected the genes based on FDR cutoff of 0.2 (for PBMC) and 0.1 (for PDAC) and top 200 (for PBMC) or 100 genes (for PDAC) per cell type. In the revised manuscript, we made sure that the information is available.**

Relevant part in the revised manuscript:

In *Application of BLADE to in silico mixture of PBMC scRNA-seq data* of Results section:

“We constructed signature matrices that capture the true mean and the standard deviation of 1,007 DEGs measured using all of 9,439 cells (top 200 DEGs with FDR < 0.2 per cell type, combined).”

In *PBMC single-cell RNA-seq data* of Methods section:

“Top 200 differentially expressed genes per cell type were identified using a two-sided Wilcoxon Rank sum test by taking a contrast between one cell type versus the rest with an FDR cutoff of 0.2.”

In *Construction of PDAC evaluation data* of Methods section

“The signature genes were selected by the top 100 DEGs from each of the ten cell types (FDR<0.1; 818 DEGs in total), followed by obtaining mean and standard deviation from the reference data.”

• **Pearson correlation should state coefficient.**

- **We changed it in the revised manuscript.**

Reviewers' Comments:

Reviewer #1:

Remarks to the Author:

The authors have answered most of the questions appropriately, however they have not included a real bulk dataset (with cell proportions according to FACS/cytometry) in the validation section (for example the dataset in Finotello, F. et al. Molecular and pharmacological modulators of the tumor immune contexture revealed by deconvolution of RNA-seq data. *Genome Med.* 11, 34 (2019).) This would be an added value for the paper.

The authors also have not mentioned in their rebuttal how the results of the experiments changed upon application of the new algorithm (after error correction).

In the abstract, I would add 'simulated' to 'evaluation using >700 (simulated!) datasets', otherwise it might be misinterpreted by readers.

In Fig 5C and 6C (per cell type performance comparison) it might be valuable to indicate (using different size or color-range of dots) how the (average) abundance of the cell type in the mixtures is.

Reviewer #2:

Remarks to the Author:

The authors have addressed most of my concerns except one of the major concerns on the use of log-likelihood as an argument for the proposed log-normal transformation. I'm not against the log-normal transformation and I believe it is a reasonable transformation, in particular for computational efficiency. However, I'm still not convinced that log-likelihood is a good metrics for picking models. As a quick example, I simulated 100 data points from a standard normal distribution and shifted them to make all data points positive, which again are normally distributed. I then calculate the maximal log-likelihoods of the original normal data and the log-transformed data. The likelihood from the log-normal model is much larger than the one from the normal model. However, the normal model is the true model. It might be more reasonable to calculate MSE of the observed and predicted count data from each model and see which model gives the smallest MSE.

Also some minor comments/questions:

1. please be consistent in citing level #s. For example, page 9 line 229, "level 1" and "level4" are cited. Either don't allow the space in "level 1" or add a space in "level4". This happens at other places too.
2. page 9 line 237, any explanation why "the performance of MuSiC gets higher from level 1 to 3". This seems counter-intuitive.
3. For the real data analysis, what is the s value used? Do the results sensitive to the choice of s ?
4. page 21, line 520, why different FDR levels used for the two datasets? Any suggestions for the choose of FDR in practice?

Reviewer #3:

Remarks to the Author:

My concerns have been addressed in the revision.

We like to thank all the reviewers for their positive response and constructive comments. We further build on our manuscript based on these comments. Specifically,

- We now included a real bulk dataset of PBMC immune cell mixtures (Finotello, F. et al.), following the suggestion. The PBMC scRNA-seq data used in our study reflect only two subjects, and it misses one of the cell types in the immune cell mixture, neutrophils. Therefore, we employed this data to evaluate BLADE for deconvolution when there is only an incomplete prior knowledge available. In this evaluation, we confirmed the superior performance of BLADE compared to other methods. Surprisingly, the other methods failed to detect many cells, particularly regulatory T cells missed in all methods other than BLADE. The robustness of BLADE is most likely thanks to the hierarchical approach of the Bayesian framework that makes it less dependent on prior knowledge.
- We also introduced an alternative metric (accuracy in detecting the most probable gene expression level) to complement log-likelihood used to compare log-normal, normal, and negative-binomial distribution. The metric is similar to mean squared error but can compare two uni-modal distributions instead of evaluating point estimates. The normal distribution is clearly the least accurate in this metric due to the bias introduced especially when the empirical distribution is skewed.

We will reply to the reviewer comments one by one.

Reviewer #1 (Remarks to the Author):

The authors have answered most of the questions appropriately, however they have not included a real bulk dataset (with cell proportions according to FACS/cytometry) in the validation section (for example the dataset in Finotello, F. et al. Molecular and pharmacological modulators of the tumor immune contexture revealed by deconvolution of RNA-seq data. *Genome Med.* 11, 34 (2019).) This would be an added value for the paper.

We appreciate this comment. Following the suggestion, we obtained the raw gene expression data generated by Finotello, F. et al. We noted some cell types not covered by our PBMC data (neutrophils and also other cell types not quantified by flow cytometry). Therefore, we positioned this data to evaluate BLADE when there is only limited prior knowledge available for the cell types that contribute to the bulk RNA-seq data. Given the same signatures provided to BLADE and other baseline methods, we noted a superior performance of BLADE compared to the other methods. The robustness of BLADE is thanks to its hierarchical Bayesian approach for integrating prior information, which makes it relatively flexible from the given prior knowledge. Though, the common failure of predicting mDC fractions in all methods indicates that the quality of prior knowledge does matter. Overall, this is a valuable extra experiment that demonstrates the robustness of BLADE, as the reviewer anticipated.

Relevant part in the revised manuscript:

- a new subsection in the *Result* section: *Application of BLADE to standard bulk RNA-seq data with incomplete prior knowledge*, and new figure (Figure 6) associated with this part.
- a new subsection in *Method* section: *Standard bulk RNA-seq data for PBMC immune cell mixtures*.

The authors also have not mentioned in their rebuttal how the results of the experiments changed upon application of the new algorithm (after error correction).

We want to apologize for not being clear enough about the changes due to the correction of the error in the code. We reproduced all the figures already based on the corrected code, but the difference was minor and thus not visible. The only difference was that the slight dip in performance with low number cell types got less apparent (see Supplementary Figure S9). The error concerns one of the parameters for modeling gene expression variability, so its influence is limited than major parameters which determine cell type fractions and cell-type-specific gene expression profiles. However, it is a very important fixation to be made for theoretical correctness, and we are happy that we were able to detect this before the final version gets published. The correction is reflected in the detailed derivation of BLADE in Supplementary Note 2.

Relevant part in the revised manuscript:

- Supplementary Note 2

In the abstract, I would add 'simulated' to 'evaluation using >700 (simulated!) datasets', otherwise it might be misinterpreted by readers.

Thanks to the above suggestion, we do have real data in the manuscript. To be completely clear that we use many simulation data in the manuscript, we revised the text to clarify that we used both simulation and real data sets.

Relevant part in the revised manuscript:

- In *Abstract*:

“Throughout an intensive evaluation with >700 simulated and real datasets, BLADE demonstrated ...”

In Fig 5C and 6C (per cell type performance comparison) it might be valuable to indicate (using different size or color-range of dots) how the (average) abundance of the cell type in the mixtures is.

The information of cell type fractions is available in Fig. 5D and Fig. 6D. Still, we agree with the reviewer that it is easier for the readers to compare fractions and deconvolution performance if this is also shown in Fig. 5C and Fig. 6C. Therefore, we indicated the cell type fractions by the size of points in Fig. 5C and Fig. 6C (now Fig. 7C in the revised manuscript). We also did the same for the same type of plots in Supplementary Figures.

Relevant part in the revised manuscript:

- Figures 5C and 7C, and Supplementary Figures S16 and S18

Reviewer #2 (Remarks to the Author):

The authors have addressed most of my concerns except one of the major concerns on the use of log-likelihood as an argument for the proposed log-normal transformation. I'm not against the log-normal transformation and I believe it is a reasonable transformation, in particular for computational efficiency. However, I'm still not convinced that log-likelihood is a good metrics for picking models. As a quick example, I simulated 100 data points from a standard normal distribution and shifted them to make all data points positive, which again are normally distributed. I then calculate the maximal log-likelihoods of the original normal data and the log-transformed data. The likelihood from the log-normal model is much larger than the one from the normal model. However, the normal model is the true model. It might be more reasonable to calculate MSE of the observed and predicted count data from each model and see which model gives the smallest MSE.

Thank you for the suggestion. MSE and log-likelihood evaluate different characteristics; the former is the accuracy of a point estimate, while the latter is the goodness of fit. In the previous round of revision, we agreed with the reviewer that these could be complementary. MSE is particularly valuable in a predictive setting (in our case, the full deconvolution model, which 'predicts' bulk RNAseq counts by the convolution over cell types). Therefore, we assessed MSE for flexible deconvolution context. We have now moved Supplementary Fig. S4 (comparing LN and NB distributions using MSE) to the main document (Fig. 2f) to give more balanced weight to the log-likelihood and MSE based evaluations.

However, MSE is less suitable to evaluate distributions when there is no prediction per sample (i.e., no point estimates). Alternatively, we include a metric that evaluates a point estimate of the mode (i.e., the gene expression level with the highest probability) like the MSE. To this end, we identified the modes from the optimized negative binomial, normal, and log-normal distribution per gene and per cell type and then assessed the distance from the true mode as estimated from the empirical distribution. This comparison is fair since all of the distribution types considered in this study are unimodal. We confirmed that normal distribution indeed identified modes less accurately than the other two distributions (Fig. 2b), which is additionally illustrated by the four genes in Fig. 2d.

Furthermore, we also noticed that there was not enough description in the Method section, so we included more details so that it can be reproduced

Relevant part in the revised manuscript:

- In *Modeling gene-expression variability by probabilistic distribution of Result* section

“To evaluate the performance of these probability distributions on gene expression variability, we assessed 1) the maximum likelihood of fitting gene expression profiles and 2) the difference between estimated and empirical modes (i.e., the most probable gene expression level; **Figs. 2a-c**). The log-normal distribution, in general, shows the best performance in per-gene maximum likelihood, followed by the negative binomial and normal distributions (**Figs. 2a,c**). In particular, we noted a biased fit of the normal distribution towards outlier observations, which led to low accuracy in identifying modes (**Fig. 2b**; see four example genes with a biased fit with normal distribution in **Fig. 2d**). In terms of mode estimation log-normal and negative binomial appear to be fairly competitive with a somewhat worse median, but a better third quartile for the former (**Fig. 2b**).”

- A new subsection in *Method* section: *Comparison between Log-normal, Negative Binomial and Normal distribution in fitting raw gene expression counts*

- In *Comparison of LN and NB based on the generic deconvolution technique of Methods* section:

“As an alternative metric, we also measured the accuracy in reconstructing bulk gene expression levels based on deconvolution. Taking actual and predicted bulk gene expression level in LN or NB deconvolution model, root mean-squared error (RMSE) was evaluated per gene and per model.”

Also some minor comments/questions:

1. please be consistent in citing level #s. For example, page 9 line 229, "level 1" and "level4" are cited. Either don't allow the space in "level 1" or add a space in "level4". This happens at other places too.

Thanks for pointing out the mistake, we made sure there is a space between after “level” in the revised manuscript.

2. page 9 line 237, any explanation why "the performance of MuSiC gets higher from level 1 to 3. This seems counter-intuitive.

It is quite an interesting outcome that we did not discuss enough in the text. When we carefully evaluate the performances, CIBERSORTx and BLADE also got better at level 3 than level 2. However, it is less obvious than MuSiC (e.g., improvement for BLADE is better visible with the Spearman correlation coefficient). One thing clear from the simulation experiment is that having more genes tends to be useful for deconvolution

(see Fig. 4 and Supplementary Fig. S5-S6). Since we identified more DEGs as the number of cell types increased, the advantage of having more genes had a more significant effect on the outcome than the extra complexity introduced by including more cell types. We extend the discussion a bit on this point.

Relevant part in the revised manuscript:

In Application of BLADE to in silico mixture of PBMC scRNA-seq data of Result section:

“Interestingly, the performance was often higher in level 3 than level 2, especially for MuSiC, most likely since the advantage of having more genes overcomes the complexity due to the increased number of cell types (e.g., 880 genes in level 3, compared to 604 genes in level 2).”

3. For the real data analysis, what is the s value used? Do the results sensitive to the choice of s ?

BLADE can indeed be sensitive to the choice of hyperparameters, including s . That is why we allow users to provide multiple possible hyperparameters and then find the best configuration using an empirical Bayes framework. Furthermore, the empirical Bayesian framework chooses the parameters without knowing true fractions and cell-type-specific gene expression levels. In this way, we can alleviate the difficulty in selecting parameters. As described in the Method section (subsection Selection of hyperparameters based on the empirical-Bayes framework), the final choice of s value is from $\{1, 0.3, 0.5\}$. Note that we kept it consistent across all the data sets used in the study.

4. page 21, line 520, why different FDR levels used for the two datasets? Any suggestions for the choose of FDR in practice?

We chose a less stringent cutoff for PBMC as there are fewer differentially expressed genes compared to the PDAC. The performance is best when there are more than 500 genes (see Supplementary Figures S5-7). Note that lowering the FDR may negatively influence performance, as there will be more genes with subtle differences in expression levels between cell types that may confuse deconvolution algorithms. We add this information in the Method section.

Relevant part in the revised manuscript:

In Construction of PDAC evaluation data of Methods section:

“ Note that we used more stringent criteria to select DEGs than for the PBMC data, because a sufficient number of DEGs (>500 DEGs) still satisfies these.”

Reviewer #3 (Remarks to the Author):

My concerns have been addressed in the revision.

Reviewers' Comments:

Reviewer #1:

Remarks to the Author:

The authors have adequately adapted the manuscript based on my comments. I have no further comments.

Reviewer #2:

Remarks to the Author:

The authors have addressed all my concerns and there are no further ones from me.